# USDC: A Dataset of User Stance and Dogmatism in Long Conversations

## Abstract

Although prior studies have explored Stance and Dogmatism in user conversations, their datasets are constructed at the post level, treating each post as independent and randomly sampling posts from conversation threads. Consequently, these datasets cannot capture users' opinion fluctuations expressed throughout the entire conversation context. Identifying these user's opinion fluctuations in long conversation threads on various topics is extremely critical for various applications, including enhanced personalization, market research, political campaigns, customer service, targeted advertising, and content moderation. Therefore, training language models to automate this task is essential. However, gathering manual annotations to train such models presents multiple challenges: 1) It is time-consuming and costly; 2) Conversation threads could be very long, increasing the chances of noisy annotations; and 3) Interpreting instances where a user changes their opinion within a conversation is difficult because often such transitions are subtle and not expressed explicitly. Inspired by the recent success of large language models (LLMs) in complex natural language processing tasks, we leverage Mistral Large and GPT-4 to automate the human annotation process on two tasks while also providing reasoning: i) User Stance classification, which involves labeling a user's stance in a post within a conversation on a five-point scale; ii) User Dogmatism classification, which involves labeling a user's overall opinion in the conversation on a four-point scale. By applying the Majority voting on zero-shot, one-shot, and few-shot annotations from these two LLMs on 764 multi-user Reddit conversations, we curate the USDC dataset. USDC is then used to finetune and instruction-tune multiple deployable small language models for the 5-class stance and 4-class dogmatism classification tasks. Additionally, human annotations on 200 test conversations achieved inter-annotator agreement scores of 0.49 for stance and 0.50 for dogmatism, indicating a reasonable level of consistency between human and LLM annotations. We make the code and dataset publicly available [1].

## 1 Introduction

Understanding fluctuations in a user's (or author's) opinions during a conversation is a fundamental to successful interpersonal interactions. It is essential for developing better communication skills, fostering empathy, and making informed decisions. This understanding is particularly relevant in the context of dogmatism—a phenomenon observed in areas such as politics, religion, culture, intellect, and science—where rigid adherence to beliefs often hinders open-mindedness and empathy (Rokeach, 1954). By aligning with the opinions and stances of potential customers, advertisers can target their campaigns more effectively. Companies can leverage this information for market research, tailoring products and services to meet consumer needs and preferences. Similarly, political groups can gauge public reactions to policies and campaigns, adjusting their strategies accordingly. Identifying differing opinions can facilitate conflict resolution by helping to understand the perspectives of all parties. By recognizing and respecting diverse opinions, society can promote tolerance and maintain social harmony.

Fig. 1 illustrates a sample Reddit conversation on the topic of *Capitalism vs. Socialism*. In this context, an author's initial post—comprising the title and body—is referred to as a submission. Multiple

---

[1] https://anonymous.4open.science/r/USDC-0F7F

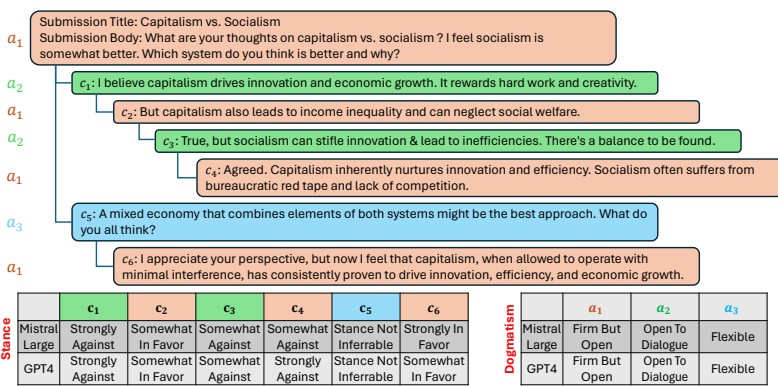

Figure 1: Sample Reddit conversation on "Capitalism vs. Socialism" with Stance (for every comment $\{c_i\}_{i=1}^6$) and Dogmatism (for every author $\{a_j\}_{j=1}^3$) labels from Mistral Large and GPT-4. The submission content favors socialism and examines how the authors position their opinions regarding socialism vs. capitalism.

authors can then share their opinions as comments on this submission. Specifically, this example contains 6 comments $\{c_i\}_{i=1}^6$ from 3 authors $\{a_j\}_{j=1}^3$. We also display stance and dogmatism predictions from two LLMs: Mistral Large and GPT-4. Some authors, like $a_1$, change their views during the discussion based on the beliefs or opinions of others. At the beginning of the dialogue, author $a_1$ somewhat favors socialism (in submission and $c_2$). However, after considering the viewpoints of author $a_2$ in comments $c_1$ and $c_3$, $a_1$ shifts their stance to somewhat favoring capitalism (in $c_4$), illustrating a firm yet open-minded approach. On the other hand, author $a_3$ seems very flexible based on their comment $c_5$.Conversely, author $a_3$ appears very flexible based on their comment $c_5$. Understanding such conversations requires comprehending the fine-grained topics being discussed and the dynamic viewpoints of individual users.

Given the importance of understanding these user dynamics in conversations, training language models to perform this task automatically at scale is critical. While several prior studies have explored Stance and Dogmatism at the post level, and numerous datasets exist for analyzing individual user posts (Fast & Horvitz, 2016; Sakketou et al., 2022; Villa-Cox et al., 2020; Li et al., 2023; Niu et al., 2024), these typically involve random subsampling or selecting posts with a limited number of tokens, treating each post as independent. Consequently, the comprehensive exploration of a specific user's opinion fluctuations within an entire conversational thread remains underexplored.

Crowdsourcing is one possible approach to address the need for a suitable dataset. However, manually annotating datasets for user opinions is time-consuming and costly, as annotators must read entire conversations to label each user's posts. Additionally, manual annotation often faces challenges related to quality, as accurately labeling opinions requires understanding demographic details and domain-specific knowledge. Given these limitations, achieving a comprehensive and accurate set of user opinions corresponding to posts about a topic often requires multiple annotators or iterative rounds of annotation. Since users can change their opinion (often with subtle transitions and not with explicit statements) within a conversation, tracking such changes across multiple users manually becomes very cumbersome.

Recently, large language models (LLMs) (Touvron et al., 2023a;b; Jiang et al., 2023; Zhang et al., 2023b), especially those built on Transformer architectures (Vaswani et al., 2017) and pretrained on large datasets, have resulted in state-of-the-art accuracies on several complex natural language processing (NLP) tasks (Brown et al., 2020; Chung et al., 2024). LLMs are also frequently used for synthetic dialog response generation (Zhang et al., 2020; Bao et al., 2019; Roller et al., 2021; Adiwardana et al., 2020). Given the complex and cumbersome nature of conversation understanding, we hypothesize that LLMs can effectively capture the nuances involved in understanding user opinions and their shifts in multi-user conversational contexts. Furthermore, since these models possess long-range memory capabilities, we believe they can reason over extended conversational threads involving numerous participants, as good as human annotators, if not better.

In this work, we leverage LLMs like Mistral Large (Jiang et al., 2023) and GPT-4 (OpenAI, 2023) to perform two tasks: i) User Stance classification, which involves labeling a user's stance of a

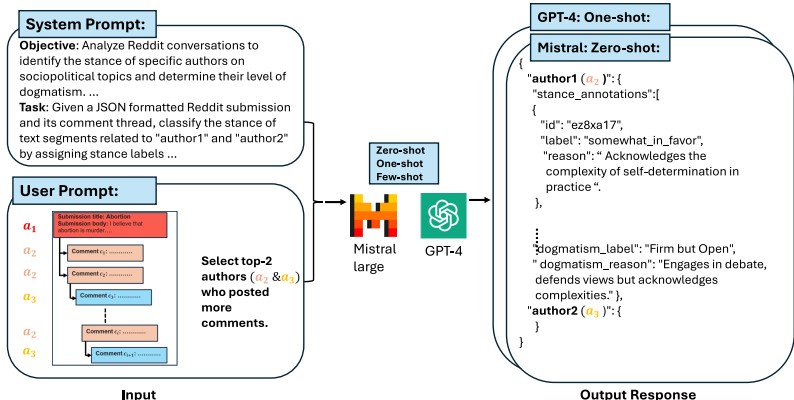

Figure 2: Generating annotations using LLMs: We pass the entire conversation for each Reddit thread in JSON format. The JSON includes the top two authors who posted the most comments, alongside annotation guidelines for stance and dogmatism labels in the system prompt.

post in a conversation on a five-point scale; ii) User Dogmatism classification, which deals with labeling a user's overall opinion in the conversation on a four-point scale. Besides the class labels, we also obtain the reasoning behind these labels from LLMs. We experiment with these two models as human-like annotators, generating user opinions in full-length, multi-user Reddit conversations in zero-shot, one-shot, and few-shot setups. Thus, for every sample, we obtain annotations in six settings ({Mistral Large, GPT-4}×{zero-shot, one-shot, few-shot}). Fig. 2 presents our LLM-based annotation pipeline for user-level Stance and Dogmatism tasks. We consider majority voting over these six settings as our final annotations. This approach enables us to curate our USDC (a dataset of user stance and dogmatism in conversations) dataset, which consists of 764 multi-user conversations from 22 subreddits, including 1,528 user-level dogmatism samples and 9,618 stance samples across all posts from selected users. The annotations in the dataset highlight specific user opinions in each post related to stance, track opinion fluctuations leading to a dogmatic nature, and provide reasoning about why users hold specific opinions.

USDC addresses several weaknesses of existing post level stance and dogmatism datasets. First, the full-length multi-user conversation aspect of USDC enables it to capture contextual and opinion shifts of multiple users. This feature allows it to serve as both an instruction-tuning user opinion dataset and an evaluation benchmark. We believe that the ability to perform instruction-tuning for user opinions at a large scale can bridge the gap between open-source and commercial user trait understanding models. Additionally, the in-context learning annotations using state-of-the-art LLMs in USDC make it a more comprehensive measure of how current LLMs understand complex tasks like capturing opinions. Further, the USDC dataset offers several use cases that extend its value in various domains, including, (i) Improving moderation tools, (ii) Analyzing public opinion dynamics, (iii) Enhancing dialogue systems and (iv) Creating dynamic contextual user representations. These aspects make it a valuable resource, especially for social media agents seeking deeper insights into user behavior.

To demonstrate the utility of USDC, we utilize our dataset to fine-tune and instruction-tune open-source LLMs for generating stance and dogmatism labels for users. We experiment with three pretrained small language models (SLMs) like LLaMA-2-7B, LLaMA-3-8B (Touvron et al., 2023b), and Falcon-7B (Almazrouei et al., 2023). We also experiment with four instruction-tuned SLMs like LLaMA-2-chat-7B, LLaMA-3-8B-instruct, Vicuna-7B-v.1.5, and Falcon-7B-instruct. Additionally, we conduct a comprehensive evaluation of the USDC dataset by incorporating human annotations on 200 test conversations and measuring inter-annotator agreement between LLM and human annotations. When comparing the LLM-generated annotations with human annotations, it becomes evident that the "lost in the middle" phenomenon (Liu et al., 2024) is marginal in LLMs, whereas human annotators maintain a steady understanding and agreement throughout the conversation, regardless of its length or complexity. Additonally, the "recency bias" phenomenon (Peysakhovich & Lerer, 2023) shows that human annotators rely heavily on the full context to maintain better inter annotator agreement with LLMs.

We make the following contributions: 1) We introduce USDC, a dataset of user stance and dogmatism in conversations dataset. 2) We benchmark initial results for the stance and dogmatism tasks using seven SLMs for the UDSC dataset. We find that stance performance improves with instruction-tuning (F1-score of 56.2) compared to finetuning (F1-score of 54.9). However, dogmatism performs worse with instruction-tuning (F1-score of 49.2) compared to finetuning (F1-score of 51.4), highlighting the complexity of this task. 3) We provide human annotations on 200 test conversations, achieving inter-annotator agreement scores of 0.49 for stance and 0.50 for dogmatism, indicating a reasonable level of consistency between human and LLM annotations. 4) We apply transfer learning by fine-tuning SLMs on the USDC dataset and assess the model's performance on existing post level stance datasets, including SPINOS, MT-CDS, and the Twitter-stance. We find that our transfer learning results are either comparable to or outperform prior studies. 5) We make the code, models and dataset publicly available[1].

## 2 RELATED WORK

**Post level stance and dogmatism.** Previous stance detection studies have primarily focused on evaluating stances within individual posts of users or through multi-party discussions on some specific topic in social media interactions (Villa-Cox et al., 2020; Sakketou et al., 2022; Li et al., 2023; Niu et al., 2024). Sakketou et al. (2022) introduced the post level Stance dataset, SPINOS, where each post is considered independently, without including submission posts for context, which affects the labeling by annotators. Recently, the MT-CSD dataset, introduced by Niu et al. (2024), addresses stance detection in multi-turn conversations with multiple targets, addressing different aspects of stance detection while the focus is on the multi-party discussions. In contrast to these two studies, Villa-Cox et al. (2020) specifically focus on extracting stances (denying vs. supporting opinions) from replies and quotes on controversial issues in Twitter conversations. Li et al. (2023) focus on target-specific stance detection, where the goal is to classify individual posts or comments into a stance class related to a specific issue, such as COVID-19 vaccination. From the above studies, we clearly observe that these works focus more on stance detection at the post level, while our work emphasizes user-level opinion fluctuations. Additionally, the prior studies are limited in scope, targeting specific issues (5 topics in (Villa-Cox et al., 2020), 1 topic in (Li et al., 2023)), whereas USDC covers a broader range of general subreddits across 22 different topics.

Similar to post level stance datasets, Fast & Horvitz (2016) predicted user dogmatism on randomly sampled Reddit posts from conversations, with each post limited to 200-300 characters. One major limitation of this work is the unavailability of a public dataset, and the treatment of each post as independent. Overall, all these prior studies contrast with the USDC dataset, which focuses on tracking user-level opinions across long, multi-user conversations, capturing the evolution of stance and dogmatism over extended discussions rather than just on a specific target issues.

**Generating annotations for NLP tasks using LLMs.** Our work also relates to a growing body of literature suggesting that LLMs can perform similarly to human annotators in labeling complex NLP tasks (Zhou et al., 2022; Zhang et al., 2023a; Bansal & Sharma, 2023; Lowmanstone et al., 2023; Wadhwa et al., 2023; Honovich et al., 2023; Zheng et al., 2024; Ye et al., 2022a; Meng et al., 2022). Several studies have explored LLM-based annotation generation in zero-shot or few-shot task settings (Ye et al., 2022a; Meng et al., 2022; Ye et al., 2022b), while others have compared pairs of language models to assess the quality of annotations generated by these LLMs (Zheng et al., 2024). However, these studies focused on generating annotations for NLP tasks such as sentiment analysis, natural language inference (Gilardi et al., 2023; Alizadeh et al., 2023), or creating synthetic dialogues, but only for dyadic conversations (Lee et al., 2023). Our approach complements these previous studies by focusing on generating annotations of user opinions in complex multi-user conversations.

## 3 USDC DATASET CURATION

### 3.1 COLLECTION OF REDDIT CONVERSATION THREADS

**Initial crawl.** We crawl a year (2019) of multi-user conversation data from 22 subreddits of Reddit using praw API [2]. This dataset includes submissions and all associated user comments. Each

---

[2]https://github.com/praw-dev/praw

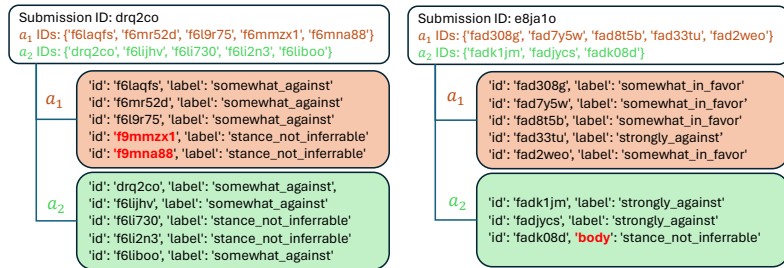

Figure 3: Failure cases of LLMs: Mistral Large few-shot output (left), the ids ("f6mmzx1","f6mna88") were mismatched with generated ids ("f9mmzx1","f9mna88"), GPT-4 zero-shot output (right), the key "label" was mismatched with generated key "body".

submission, which serves as the initial message of the conversation, contains a title and content body. This is followed by comments and replies to the submission or other comments. Overall, we crawled 3,619 Reddit conversations across the 22 subreddits. A sample Reddit conversation is shown in Fig. 1.

**Quality filtering of conversations.** Since submission content on Reddit can sometimes include videos, we perform the following filtering steps. 1) We only consider submissions where the content is text. 2) We remove conversations with [deleted] tags and empty content. 3) We exclude conversations where the posts were discarded by users or removed by moderators.

Reddit user conversations can be very long, and we observed up to 591 comments in a single crawled conversation data. Considering the maximum sequence length allowed by various language models, we retained only those conversations that contain at least 20 and, at most, 70 comments, as shorter conversations (fewer than 20 comments) are insufficient for accurately gauging user opinions. Further, we ensure that at least two users covering ∼50% of the comments in the conversations. We did not remove any comments or reduce the post length in the selected conversations. Out of the initial 3,619 conversations, these filtering steps result into 764 conversations getting selected. Table. 3 in the Appendix B shows detailed subreddit level statistics.

## 3.2 Obtaining LLM Annotations

**Representing Reddit conversations in JSON format.** To create the prompt, we follow the nested hierarchical structure of Reddit conversations to maintain the context. Specifically, we maintain a JSON structure for each conversation, where each author has their post IDs, and comments or replies are available in the body section. An example of a Reddit conversation in JSON format is provided in Appendix E. Note that the JSON explicitly includes the top-2 authors who posted the most comments in the conversation, and their respective post IDs. Our emphasis on these top-2 users (covering 47% posts of total posts on average) aimed at accurately assigning Stance and Dogmatism labels, acknowledging the challenge of modeling a user's opinion belief based on a very limited number of posts within a conversation.

**Using LLMs as human-like annotators.** To annotate the stance of a user towards a submission at each individual post and to assess the overall level of dogmatism expressed by the user throughout the conversation, we employ two well-known commercialized API-based LLMs: GPT-4 (OpenAI, 2023) and Mistral Large (Jiang et al., 2024). OpenAI GPT-4 is a decoder-based language model with a context window of 32k to 128k tokens. Mistral Large features a context window of 32k tokens. Additionally, we examined other versions of these models, such as GPT-3.5 and Mistral-small and medium, but found that these models failed to produce annotations in the desired format. We briefly discuss these limitations, along with the situations where LLMs are prone to errors, in Appendix F.

For both GPT-4 and Mistral Large, we supplied a system prompt that contains the definitions of Stance and Dogmatism, guidelines for annotating each user conversation, and the necessary labels for Stance and Dogmatism, as shown in Fig 2. The system prompt is detailed in the Appendix C. Along with the system prompt, we provided a user prompt comprising the entire user conversation in a structured JSON format, as discussed above. Additionally, we prompted the model to generate reasoning for each label, explaining why the LLMs assigned a particular label to a specific user post. We used zero-shot, one-shot, and few-shot settings to get the LLM-based annotations. For

the few-shot setting, we added two examples in the prompt. Samples of generated outputs using GPT-4 in zero-shot, one-shot, and few-shot settings are shown in Appendix G.1, G.2, G.3 respectively. Similarly, samples of generated outputs using Mistral Large in zero, one, and few-shot settings are shown in Appendix G.4, G.5, G.6 respectively.

**Annotation tasks.** We prompt the LLMs to perform two annotation tasks: 1) Stance detection, which determines if a user comment or post is *Strongly In Favor*, *Strongly Against*, *Stance Not Inferrable*, *Somewhat In Favor*, or *Somewhat Against* towards specific subreddit submission content; Our 5-class stance detection scheme is inspired by the SPINOS dataset proposed by Sakketou et al. (2022). These labels provide a fine-grained analysis similar to sentiment labels, allowing for a more detailed understanding of user opinions. 2) Dogmatism identification, which evaluates the user's overall opinion in conversation and categorizes them into one of four categories: *Firm but Open*, *Open to Dialogue*, *Flexible* or *Deeply Rooted*. Our 4-class dogmatism task is inspired by Fast & Horvitz (2016), where the authors reported ratings that correspond to each level of dogmatism. We have adopted similar definitions for dogmatism labels and incorporated them into our system prompts to ensure consistency and accuracy in our annotations. This assessment reveals whether users are open to changing their beliefs or remain steadfast in their opinions based on interactions with other users.

**Addressing failures in JSON parsing of LLM response.** Sometimes, LLMs get confused with the author IDs and miss their Stance labels (Fig. 3 (left)). Sometimes, there were minor errors in key naming ('label' vs 'body' in Fig. 3 (right)). We observed such errors in ∼15 cases across LLM setting. We manually fixed JSON parsing errors and corrected author IDs for associated Stance labels.

**Majority voting conflict.** After obtaining six annotations ({Mistral Large, GPT-4}×{zero, one, and few-shot}) for each sample, we follow the two step process to obtain final gold annotations. (i) Majority voting: we aggregate using majority voting (i.e label that appears most frequently across models) to determine the final gold annotations for the Stance and Dogmatism tasks. (ii) Handling situations with no clear majority: when generating annotations using both GPT-4 and Mistral Large, it is possible that the two models might provide different annotations for the same conversation. In these cases, we use the annotation provided by GPT-4 in the few-shot setting as the deciding factor or "gold standard". We chose to prioritize GPT-4 few-shot annotations because human annotations have better IAA agreement with GPT-4 few-shot. Further, few-shot models, which are fine-tuned with a small amount of task-specific data, often provide more accurate and contextually relevant annotations.

Class distributions for stance task is as follows: 3117 (somewhat in favour), 2266 (stance not inferrable), 1998 (somewhat against), 1303 (strongly against) and 640 (strongly in favor). For dogmatism task, the distribution is as follows: 666 (open to dialogue), 653 (firm but open), 140 (deeply rooted), and 69 (flexible). We present the class distributions obtained from each model with the 3 settings (zero, one, and few-shot) for both the tasks in Figs. 4 and 5, respectively, in Appendix B.

### 3.3 Inter-annotator Agreement with LLMs as Annotators

As the quality of labeling on subjective tasks is challenging, we validate the inter-annotator agreement (IAA) between the two LLMs in three settings (GPT-4 Zero-shot, GPT-4 One-shot, GPT-4 Few-shot, Mistral Large Zero-shot, Mistral Large One-shot, and Mistral Large Few-shot) for the Stance and Dogmatism tasks. We perform IAA using two approaches: i) Cohen's kappa score (Cohen, 1960) and ii) Fleiss' kappa score (Fleiss, 1971). Cohen's kappa measures the agreement between two raters, while Fleiss' kappa extends this to multiple raters. Hence, we employed Cohen's kappa for pairwise comparisons and Fleiss' kappa for overall agreement across all models.

Fig. 6 in the Appendix B shows pairwise Cohen's kappa values for both tasks. We observe that Cohen's kappa values range from 0.36 to 0.72 for stance and 0.31 to 0.61 for dogmatism, indicating moderate agreement between the models. Broadly, kappa values are higher for model pairs within a family (GPT-4 or Mistral large). Thus, the large variance in the kappa scores is not due to the various in-context learning settings (ZS, OS, FS) but rather due to architectural differences.

The overall Fleiss' kappa value was calculated as 0.485 for stance and 0.435 for dogmatism, suggesting moderate agreement among all six settings. Comparing LLM IAA with previous studies, we observe that for dogmatism, the LLM IAA of 0.435 matches with 0.44 as mentioned in (Fast & Horvitz, 2016). Similarly, for Stance, the LLM IAA of 0.485 is much higher than 0.34 as reported in (Sakketou et al., 2022). This suggests that LLMs can be considered as competent annotators for complex subjective tasks.

## 3.4 USDC TEST DATASET EVALUATION WITH HUMAN LABELS

Due to the time-consuming nature of the manual annotation process, we perform human annotations on a set of 200 test conversations. In the forms for human annotations, we displayed the top 2 authors Reddit posts from the conversation, along with the submission title and content. We also provided a link to the original Reddit URL so that annotators could look at the full conversation. We provided detailed annotation guidelines (similar to the ones mentioned in the prompt in Appendix C) to instruct human annotators in carrying out these tasks.

With three human annotators on a sample of 200 conversations, as shown in Appendix O Fig. 20, we achieved an inter-annotator agreement score of 0.49 for the stance detection and 0.50 for dogmatism tasks, indicating a reasonable level of consistency between human and LLM annotations. The annotators included two males and one female, affiliated with academia and industry, aged between 20 and 40, who were very familiar with Reddit topics. We calculated the inter-annotator agreement among the three human annotators themselves. Tables 12 and 13 in Appendix P report the IAA scores for both stance and dogmatism tasks among the human annotators. The results showed an agreement of 0.57 for the stance and 0.52 for the dogmatism. These findings demonstrate the level of consistency among human annotators, providing a more comprehensive understanding of the alignment between LLM-generated labels and human judgments.

## 4 TRAINING SMALL LANGUAGE MODELS (SLMS)

In this section, we briefly discuss the small language models that we experiment with. We also discuss their finetuning and instruction-tuning details. We train three pretrained small language models (LLaMA-2-7B, LLaMA-3-8B, Falcon-7B) and four instruction-tuned small language models (LLaMA-2-chat-7B, LLaMA-3-8B-instruct, Vicuna-7B-v.1.5, and Falcon-7B-instruct). We finetune and instruction-tune these models using the proposed USDC dataset. We use pretrained model checkpoints from Hugging Face (Wolf et al., 2020). All of these LLMs have a context length of 4096 tokens. Model details and hyper-parameter settings are in Appendix H.

**Train-test setup.** We conducted both finetuning and instruction-tuning of small language models. For this purpose, we divided the dataset of 764 conversations into train ($\sim 75\%$) and test splits ($\sim 25\%$). The training dataset comprised 564 conversations, including 1128 samples of dogmatism labels and 7520 samples of stance labels. Conversely, the testing dataset consisted of 200 conversations, with 400 samples of dogmatism labels and 1831 samples of stance labels across two authors posts.

**Finetuning of SLMs.** For Stance classification, we treat each user post as an independent sample. In contrast, for the dogmatism classification, we consider the entire user conversation as a single sample by concatenating all the threads from a user in that conversation. To load the pretrained SLMs, we perform 4-bit quantization, and we finetune the models by apply the LoRA technique (Hu et al., 2021), with SFTT before saving the finetuned model. For finetuning, we used prompt for Stance classification as shown in Fig. 7 (see Appendix D). Similarly, Fig. 8 (see Appendix D) displays prompt for Dogmatism identification.

**Instruction-tuning of SLMs.** We instruction-tune the SLMs on user conversations along with their gold labels from the training part of the USDC dataset. For instruction-tuning, we use the same prompt as used for LLMs to generate the USDC dataset (also shown in Appendix C). Similar to finetuning, we use same train-test splits for instruction-tuning.

## 5 RESULTS

**Baseline (un-fine-tuned) model performance and what constitutes a "reasonable" F1 score?**
To establish a reasonable F1-score benchmark for fine-tuning and instruction-tuning (discussed in the next subsections), we evaluated the un-fine-tuned SLMs, GPT-4 and Mistral Large, in few-shot settings. This evaluation includes both stance and dogmatism tasks, using majority voting to enhance reliability. The results are summarized in the Tables 4, 5, 6 and 7 in Appendix I. We make the following observations: (i) Majority Voting generally provides a slight improvement over individual few-shot configurations, which suggests the value of combining predictions from multiple models. (ii) The difference between GPT-4 and Mistral Large in un-fine-tuned few-shot settings is relatively

Table 1: Finetuning and instruction-tuning results: weighted F1-score for Stance and Dogmatism classification using SLMs on USDC test set. ZS: Zero-shot, OS: One-shot, FS: Few-shot.

| | Model | Stance Classification | | | | | | | Dogmatism Classification | | | | | | |
|---|---|---|---|---|---|---|---|---|---|---|---|---|---|---|---|
| | | GPT-4 | | | Mistral Large | | | Majority | GPT-4 | | | Mistral Large | | | Majority |
| | | ZS | OS | FS | ZS | OS | FS | | ZS | OS | FS | ZS | OS | FS | |
| Finetuning | LLaMA-2-7B | 51.8 | 52.9 | 52.7 | 35.1 | 49.2 | 46.0 | 54.0 | 42.1 | 44.2 | 45.2 | 39.3 | 47.6 | 43.7 | 43.4 |
| | LLaMA-2-chat-7B | 52.8 | 51.4 | 51.8 | 34.7 | 47.5 | 46.5 | 51.3 | 42.1 | 42.5 | 48.8 | 41.1 | 49.7 | 45.5 | 48.3 |
| | LLaMA-3-8B | 51.3 | 52.2 | 52.9 | 34.9 | 48.5 | 47.0 | **54.9** | 42.0 | 47.8 | 45.3 | 39.9 | 47.4 | 36.3 | **51.4** |
| | LLaMA-3-8B-instruct | 51.2 | 52.6 | 52.7 | 33.9 | 49.5 | 45.6 | 54.5 | 44.8 | 46.2 | 49.7 | 46.1 | 45.8 | 46.1 | 50.8 |
| | Falcon-7B | 50.7 | 51.1 | 51.6 | 34.9 | 47.2 | 43.9 | 53.2 | 41.5 | 42.1 | 43.3 | 36.5 | 38.4 | 37.5 | 40.1 |
| | Falcon-7B-instruct | 51.2 | 51.5 | 51.6 | 35.1 | 47.7 | 44.2 | 51.0 | 41.7 | 42.1 | 42.9 | 36.8 | 38.5 | 36.9 | 39.7 |
| | Vicuna-7B-v.1.5 | 51.0 | 53.0 | 53.2 | 35.1 | 48.3 | 45.8 | 54.7 | 42.9 | 48.3 | 40.8 | 45.9 | 42.6 | 46.2 | 42.3 |
| Instruction-tuning | LLaMA-2-7B | 53.2 | 54.0 | 54.5 | 36.8 | 50.3 | 47.2 | 55.5 | 43.0 | 45.0 | 46.3 | 40.6 | 48.2 | 45.0 | 44.0 |
| | LLaMA-2-chat-7B | 54.0 | 54.5 | 55.0 | 36.5 | 50.7 | 47.6 | 54.0 | 43.2 | 45.5 | 47.0 | 40.8 | 48.5 | 45.5 | 43.8 |
| | LLaMA-3-8B | 53.5 | 54.8 | 55.5 | 37.0 | 50.5 | 48.0 | **56.2** | 43.5 | 46.0 | 47.5 | 41.0 | 48.8 | 45.8 | 45.1 |
| | LLaMA-3-8B-instruct | 53.0 | 54.2 | 55.0 | 36.0 | 50.0 | 47.0 | 55.5 | 43.8 | 46.5 | 47.8 | 41.5 | **49.2** | 46.0 | 44.8 |
| | Falcon-7B | 52.8 | 53.4 | 54.0 | 36.5 | 49.5 | 46.5 | 54.8 | 42.5 | 44.6 | 45.8 | 39.8 | 47.0 | 44.0 | 43.8 |
| | Falcon-7B-instruct | 53.0 | 53.8 | 54.2 | 36.8 | 49.8 | 46.8 | 54.5 | 42.8 | 44.8 | 46.0 | 40.0 | 47.2 | 44.2 | 43.0 |
| | Vicuna-7B-v.1.5 | 53.3 | 54.5 | 55.2 | 37.0 | 50.2 | 47.8 | 55.2 | 43.7 | 46.8 | 47.2 | 41.2 | 48.2 | 46.5 | 44.8 |

small, indicating that both models are fairly comparable in performance on these tasks when using the LLaMa-3-8B model.

As shown by the un-fine-tuned model's performance for stance classification in Table 6, an overall accuracy of 0.311 and F1 scores as low as 0.06 for certain classes, the baseline for this task is relatively low. Similarly, for dogmatism in Table 7, an overall accuracy of 0.40 and F1 scores as low as 0.00 for certain classes. In this context, an F1 score that significantly improves upon this baseline—especially if it approaches or exceeds 50%—could be considered reasonable.

**Do SLMs finetuned with task-specific LLM annotations accurately perform Stance and Dogmatism tasks on user opinions?**

We show the weighted F1 of various SLMs finetuned with task-specific LLM annotations on the stance and dogmatism detection tasks on the USDC test set in Table 1. We report AUC scores and other qualitative analysis in Appendix J (Fig. 10 and 11). We make the following observations from these results: 1) Compared to the baseline, while the un-fine-tuned models show moderate performance, the fine-tuned models nearly double their F1 scores, particularly for the Stance task. Even for dogmatism tasks, we saw better improvement in F1-score after fine tuning. 2) For both tasks when finetuning, the majority voting labels as ground truth has a relatively high performance, scoring above 50% weighted F1-score across several (7/7 for stance and 2/7 for dogmatism) models. 3) Finetuned LLaMa-3 models (LLaMA-3-8B and LLaMA-3-8B-instruct) perform better across both tasks. 4) For GPT-4 annotations, in most cases, SLMs finetuned with few-shot annotations outperform those trained with zero and one-shot annotations. For Mistral Large annotations, SLMs finetuned with one-shot annotations perform the best. 5) Specifically, for the stance detection task, Vicuna-7B-v.1.5 finetuned using few-shot annotations is the best model trained with GPT-4 annotations. Similarly, LLaMA-3-8B-instruct finetuned with one-shot annotations is the best model trained with Mistral Large annotations. 6) For the dogmatism detection task, LLaMA-3-8B-instruct finetuned using few-shot annotations is the best model trained with GPT-4 annotations. Similarly, LLaMA-2-chat-7B finetuned with one-shot annotations is the best model trained with Mistral Large annotations. 7) Overall, we observe that instruction-tuned SLMs perform better than the pretrained SLMs.

**Do SLMs instruction-tuned with task-specific LLM annotations perform better than SLMs finetuned with task-specific LLM annotations for the Stance and Dogmatism tasks?** We show the weighted F1 of various SLMs instruction-tuned with task-specific LLM annotations on the stance and dogmatism detection tasks on the USDC test set in Table 1. We report AUC scores and other qualitative analysis in Appendix K (see Fig. 12). We make the following observations from these results: 1) SLMs with instruction-tuning result in higher weighted F1-scores than SLMs with finetuning for stance detection, while SLMs with finetuning outperform SLMs with instruction-tuning in dogmatism detection. 2) Contrary to finetuning results, instruction-tuning results demonstrate that using majority voting labels as ground truth, SLM instruction-tuning yields relatively high performance only for the stance detection task, but not for the dogmatism detection. 3) Similar to finetuning results, LLaMA-3 models (LLaMA-3-8B and LLaMA-3-8B-instruct) perform better across both tasks. Additionally, GPT-4 annotations yield the best results in the few-shot setting, while Mistral Large annotations perform best in the one-shot setting.

Overall, we draw the following conclusions when comparing SLM finetuning and instruction-tuning: (1) Since dogmatism detection is inherently a more complex and varied than stance detection, the model might struggle to generalize from the instructional data. (2) The system prompt used in finetuning is much simpler than the original system prompt for instruction-tuning, making it challenging to handle the context length for longer conversations. We perform an error analysis to further analyze the results in the next subsection.

QUALITATIVE ANALYSIS

**Error Analysis.** Table 2 illustrates the confusion matrix for stance detection for LLaMa-3-8B finetuning and instruction-tuning. We make the following observations from this table: 1) For both finetuning and instruction-tuning, there is a significant misclassification between "Somewhat Against" and "Somewhat In Favor," as well as between "Somewhat In Favor" and "Stance Not Inferrable." These overlaps suggest challenges distinguishing moderate stances, indicating a need for enhanced feature representation and clearer class definitions to improve model performance. We report the confusion matrix for dogmatism detection task in Fig. 9 in the Appendix. It shows significant misclassifications, especially for the "Deeply Rooted" and "Flexible" labels, with zero accuracy and F1-scores. On the other hand, the model performs moderately better for "Firm but Open" and "Open to Dialogue" classes with accuracies of 48.7% and 64.4%, respectively. The confusion matrix also indicates substantial confusion to distinguish between intermediate levels of dogmatism, such as "Firm but Open" and "Open to Dialogue". The area under the ROC curve (AUC) measures the model's ability to distinguish between classes. Hence, we further report the ROC curve, which shows the trade-off between the true positive rate (TPR) and false positive rate (FPR) for each class for stance and dogmatism tasks, see Figs. 10 and. 11 in Appendix J.

**Lost in the Middle.** To analyze the "lost in the middle" (Liu et al., 2024) phenomenon in our LLM-based user-stance annotations, for a given user, we divided the data into time segments and calculated inter-annotator agreement (IAA) using Cohen's Kappa scores across different models and settings. The data was segmented based on the submission_id, author_id, and stance_id_timestamp. For each group (i.e., each combination of submission_id and author_id), the timestamps were divided into equal segments. The number of entries for each group was divided by the desired number of segments (3), and the division was done as evenly as possible, with each segment containing a roughly equal number of time-stamped entries. Fig. 13 in Appendix reports the comparison statistics of IAA scores for the stance detection task across initial, middle, and later time stamps. From Fig. 13, we observe that the analysis across different time segments, especially when divided into three segments, clearly demonstrates that the "lost in the middle" phenomenon is marginal.

The partial decrease in inter-annotator agreement during the middle parts of the conversations suggests that as conversations progress, models might face challenges in maintaining consistent agreement; however, the decrease in agreement scores is minimal. The recovery in agreement towards the final segments could indicate that as conversations start to conclude, they become more focused, or that the models are better able to align on concluding statements. This trend underscores the importance of considering segment-based analysis when evaluating model performance over long-form conversations. When comparing the model-generated annotations with human annotations, it becomes evident that we do not encounter the "lost in the middle" problem. The human annotations demonstrate a consistent level of inter-annotator agreement (IAA) across all three segments—initial, middle, and final. This suggests that human annotators maintain a steady understanding and agreement throughout the conversation, regardless of its length or complexity.

**Recency Bias Phenomenon (Prior Context vs. Full Context).** To investigate the impact of recency bias (Peysakhovich & Lerer, 2023) on LLM performance in user-stance annotations, we focused on verifying model annotations by examining the prior context for a given user, rather than considering the entire conversation. The goal was to determine whether assessing each response within its immediate context, followed by aggregation, would yield different results compared to analyzing the full conversation context. Further details about the prior context annotations using LLMs are discussed in Appendix O. Fig. 14 in the Appendix reports IAA scores, which contains a matrix of Cohen's Kappa scores across different models and settings, including GPT-4 Few-Shot (FS), Mistral Large FS, Majority Voting, as well as GPT-4 FS PC and Mistral Large FS PC (here, PC denotes prior context). From the figure, we observe that The agreement between GPT-4 FS and Majority Voting is higher when the full conversation is considered (0.75) compared to when only prior context is used.

Table 2: Confusion matrix for LLaMa-3-8B Stance detection models on USDC test set: finetuning (left) and instruction-tuning (right). SOA: Somewhat Against, SOIF: Somewhat In Favor, SNI: Stance Not Inferrable, SGA: Strongly Against, SIF: Strongly In Favor.

| | | Predicted | | | | | | | | Predicted | | | | |
|---|---|---|---|---|---|---|---|---|---|---|---|---|---|---|
| | | SOA | SOIF | SNI | SGA | SIF | | | | SOA | SOIF | SNI | SGA | SIF |
| | SOA | 151 | 132 | 34 | 44 | 2 | | | SOA | 143 | 125 | 37 | 54 | 4 |
| | SOIF | 93 | 537 | 113 | 17 | 14 | | | SOIF | 82 | 543 | 106 | 27 | 16 |
| Actual | SNI | 23 | 78 | 259 | 5 | 0 | | Actual | SNI | 22 | 82 | 253 | 6 | 2 |
| | SGA | 52 | 35 | 13 | 115 | 17 | | | SGA | 41 | 35 | 11 | 131 | 14 |
| | SIF | 18 | 50 | 12 | 25 | 27 | | | SIF | 16 | 53 | 10 | 23 | 30 |

The agreement between GPT-4 FS PC and Mistral Large FS PC (both based on prior context) is lower than when using the full context, indicating that prior context alone may not capture all the necessary nuances for consistent annotation.

**Human Agreement.** The comparison of human annotations with models like GPT-4 FS and Mistral Large FS shows that human annotators also rely heavily on the full conversation context to maintain agreement. The results from this additional experiment, supported by the data in Fig. 14 in Appendix, suggest that while prior context can provide some useful insights, it is not as effective as considering the entire conversation context for maintaining high inter-annotator agreement. In summary, the experiment highlights the importance of full context in LLM-based annotations and suggests that while recency can influence model performance, it should be supplemented with the entire conversation context to ensure higher accuracy and agreement.

**Transfer Learning Evaluation of Models trained on USDC.** To evaluate the quality of LLM-generated annotations, the annotators labeled 200 conversations and transfer learning is applied by fine-tuning the SLMs on the USDC dataset. We subsequently tested the model's performance on several existing stance datasets, including SPINOS (Sakketou et al., 2022), MT-CDS (Niu et al., 2024), and the Twitter stance dataset (Villa-Cox et al., 2020). We observe that performance of models trained using USDC is better or comparable to that of models trained using individual datasets themselves. Detailed results and analysis of results for the three datasets are reported in Appendix N.

# 6 DISCUSSION & CONCLUSION

We introduced USDC, a large-scale dataset of user stance and dogmatism in conversations, leveraging LLMs as human-like annotators. This dataset is used for various applications, including analyzing public opinions, enhancing dialogue systems, improving content moderation tools by identifying and flagging dogmatic or polarizing users in online discussions, and generating dynamic contextual user representations. The full-length multi-user conversation aspect of USDC allows it to capture the contextual and opinion shifts of multiple users in a conversation. We believe that the ability to perform finetuning or instruction-tuning SLMs for user opinions at a large scale can bridge the gap between SLMs and commercial LLMs for understanding user traits. While finetuning SLMs shows good F1-score on both stance and dogmatism tasks, the F1-score remains below 60% (54.9% for stance and 51.4% for dogmatism). On the other hand, instruction-tuning of SLMs only improves F1-score performance on stance, not the dogmatism task. Further, the performance still falls short of 60%, with weighted F1-scores of 56.2% for stance and 49.2% for dogmatism. These findings indicate that there is still significant room for improvement in understanding user opinions from a text segment. Human evaluation showed an agreement of 0.57 for the stance and 0.52 for the dogmatism tasks between LLM and human annotations. This indicates that LLM-generated annotations in USDC are close to human labels. Transfer-learning on 3 datasets also showed positive results.

**Limitations.** We plan to extend this work along the following directions in the future. 1) We would like to extend this work to multi-lingual conversations and verify how accurately SLMs and LLMs perform on the stance and dogmatism tasks in the multi-lingual scenario. 2) We analyzed user dogmatism based on their posts within a single conversation. This approach could be extended to include posts across multiple conversations and utilize similar profile information if available. 3) We analyzed dogmatism information for only the top two authors. Users with fewer comments often do not provide enough information to accurately assess their stance or dogmatism, as many contribute only one or two comments, which is insufficient to determine their overall opinion or dogmatic nature. Therefore, our study prioritizes the two most active users, who contribute approximately 50% of the comments in each conversation, to better capture opinion fluctuations and provide a more robust analysis of stance and dogmatism.

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

## A   OVERVIEW OF APPENDIX SECTIONS

- Section B: Detailed Statistics of the USDC Dataset
- Section C: System Prompt for LLM Annotation
- Section D: Prompts for Finetuning SLMs
- Section E: Sample of User Input Prompt
- Section F: Situations Leading to LLM Annotation Errors and Inconsistencies
- Section G: Samples of JSON Outputs from LLMs
- Section H: Details of small language models and Hyper-parameter settings
- Section I: Baseline (un-fine-tuned) model performance
- Section J: SLM Finetuning: AUC (Area Under the Curve) Analysis
- Section K: SLM instruction-tuning: AUC (Area Under the Curve) analysis
- Section L: "Lost in the Middle" Analysis
- Section M: Recency Bias Analysis
- Section N: SLM finetuning: Transfer Learning Performance
- Section O: Individual user responses within their specific context vs. entire conversation at once for stance and dogmatism
- Section P: Inter-Annotator Agreement (IAA) between human annotators
- Section Q Robustness analysis of Human-LLM Annotations
- Section R Qualitative examples demonstrating cases with high, moderate, and low inter-annotator agreement (IAA)
- Section S: Wighted Cohen's Kappa score: IAA between human labels and LLM-generated labels

## B   DETAILED STATISTICS OF THE USDC DATASET

Table 3 shows the detailed statistics of our USDC dataset at the subreddit level. Fig. 4 shows the distribution of stance labels across LLM annotations across zero-shot, one-shot, and few-shot settings. Fig. 5 shows the distribution of dogmatism labels across LLM annotations across zero-shot, one-shot, and few-shot settings.

Table 3: Statistics of the User Conversation Dataset.

| subreddit | num_conversations | min_total_token_count | max_total_token_count |
|---|---|---|---|
| DebateCommunism | 73 | 529 | 11557 |
| Abortiondebate | 70 | 1271 | 7401 |
| CapitalismVSocialism | 61 | 665 | 16927 |
| prochoice | 60 | 582 | 7278 |
| brexit | 56 | 637 | 4553 |
| climateskeptics | 56 | 734 | 7550 |
| prolife | 54 | 672 | 13342 |
| gunpolitics | 52 | 683 | 7889 |
| MensRights | 52 | 623 | 5774 |
| climatechange | 49 | 520 | 7427 |
| nuclear | 41 | 572 | 5282 |
| progun | 39 | 436 | 3632 |
| NuclearPower | 23 | 629 | 4589 |
| Vegetarianism | 22 | 627 | 3958 |
| AntiVegan | 20 | 351 | 5052 |
| climate | 13 | 701 | 4678 |
| Egalitarianism | 10 | 665 | 4060 |
| VeganActivism | 8 | 460 | 3685 |
| Veganism | 2 | 1332 | 1738 |
| AnimalRights | 1 | 845 | 845 |
| animalwelfare | 1 | 1363 | 1363 |
| GunsAreCool | 1 | 2945 | 2945 |

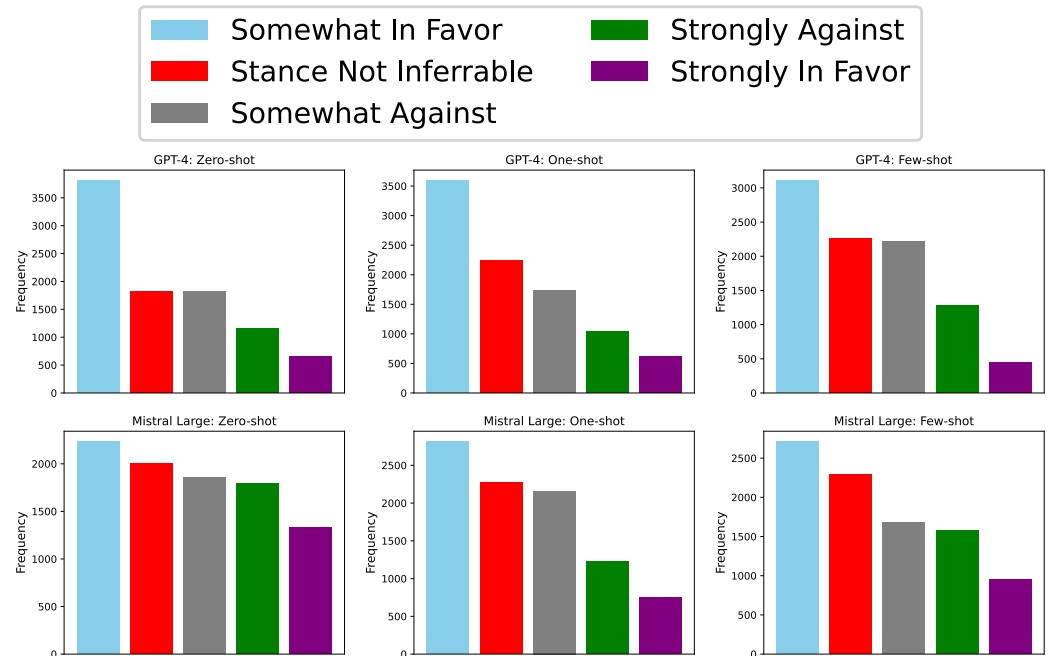

Figure 4: Distribution of Stance labels across LLM annotations in six settings: GPT-4, Mistral Large×Zero-shot, One-shot, Few-shot. Somewhat In Favor is the most frequent class across all six settings, while Strongly In Favor is the least frequent.

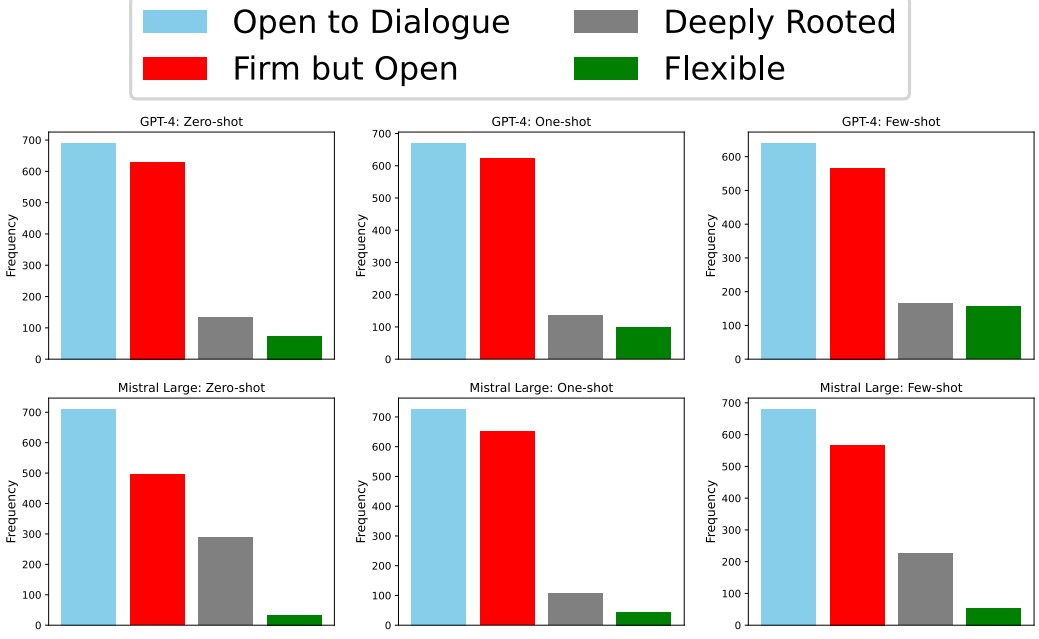

Figure 5: Distribution of dogmatism labels across LLM annotations in six settings: GPT-4, Mistral Large×Zero-shot, One-shot, Few-shot. Open to Dialogue is the most frequent class across all six settings, while Flexible is the least frequent.

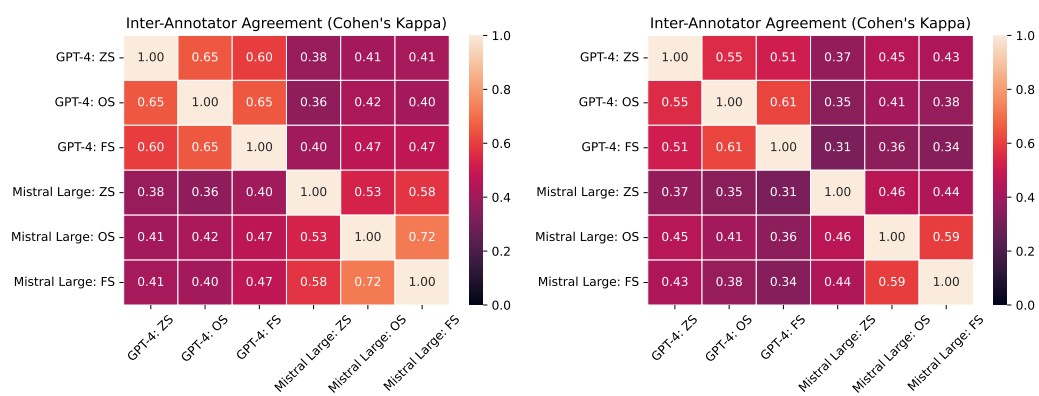

Figure 6: Inter-annotator agreement (IAA): Cohen's Kappa score across six different settings (2 models×3 settings) for Stance (left) and Dogmatism (right) tasks.

## C System Prompt for LLM Annotation

We used the following system prompt as annotation guidelines both to obtain annotations from LLMs and for the instruction-tuning of SLMs.

```
"""
### Introduction
**Objective**: Analyze Reddit conversations to identify the stance
    of specific authors on sociopolitical topics and determine their
    level of dogmatism.
**Stance Definition**: Stance is defined as the expression of the
    author's standpoint and judgement towards a given topic.
**Dogmatism Definition**: Dogmatism is an opinion strongly believed
    as a fact to support a stance without a question or allowance
    for conversation.
**Task**: Given a JSON formatted Reddit submission and its comment
    thread, classify the stance of text segments related to
    ``author1'' and ``author2'' by assigning one of the following
    five predefined stance labels: 'strongly_against',
    'somewhat_against', 'somewhat_in_favor', 'strongly_in_favor',
    'stance_not_inferrable'. Also, assign a dogmatism label for each
    author by assigning one of the following four predefined labels:
    'Deeply Rooted', 'Firm but Open', 'Open to Dialogue', 'Flexible'.

### Description of Stance Labels:
1. **strongly_against / strongly_in_favor**: Marks text showing
    strong opinions, emotional expressions, or argumentative tones.
2. **somewhat_against / somewhat_in_favor**: Identifies texts with
    openness to discussion, less certainty, or showing interest in
    different viewpoints.
3. **stance_not_inferrable**: Use for texts that are neutral,
    support both stances, or where the stance is unclear despite
    being on-topic.

### Description of Dogmatism Labels:
1. **Deeply Rooted**: Reflects a strong, unchangeable belief. This
    label conveys the idea of someone who is firm in their opinion
    and unlikely to be swayed.
2. **Firm but Open**: Indicates a person who is not likely to
    change their mind but does not impose their views
    authoritatively. It captures the essence of being steadfast in
    one's beliefs without being dismissive of others.
3. **Open to Dialogue**: Describes someone who holds a certain
    opinion but is genuinely interested in considering other
```

```
      viewpoints. This label suggests a willingness to engage in
      meaningful conversation about differing perspectives.
4. **Flexible**: Denotes a person who is not firmly committed to
   their stance and is open to changing their opinion. This label
   is indicative of flexibility and openness to new information or
   arguments.

### Input Data Format
The input data will be in JSON format and will include several key
    elements to represent a Reddit submission and its associated
    comments. Each element provides specific information as
    described below:

- 'id': This is the unique identifier for the Reddit submission.
- 'title': The title of the post. This is what users see first and
    often summarizes or hints at the content of the submission.
- 'content': The main post's detailed description. This text
    segment provides the core message or information the author
    wishes to communicate with the Reddit community. It may include
    narratives, questions, or any information relevant to the title.
- 'comments': An array (list) of comments related to the Reddit
    submission. Each comment in this array includes the following
    fields:
  - 'id': The unique identifier for the comment, allowing for
      identification and reference within the dataset.
  - 'author1' or 'author2': The username of the comment's author,
      if it is made by one of our focus authors. This helps in
      tracking contributions by specific individuals.
  - 'body': The text of the comment. This is the main content of
      the comment where the author responds to the post or another
      comment, providing insights, opinions, or further information.
  - 'replies': An array of comments that are direct responses to
      this comment. The structure of each reply follows the same
      format as the initial comment, including 'id', 'author1' or
      'author2' (if applicable), 'body', and potentially more
      'replies'.

### Output Data Format
Submit your annotations in JSON format, grouping all stance
    annotations under the key ''stance_annotations''. Each entry
    should be a dictionary containing the segment's ''id'', your
    ''label'', and the ''reason'' for your choice. Include the
    dogmatism label and its justification under ''dogmatism_label''
    and ''dogmatism_reason'' keys, respectively.

The output should follow this structure:
```json
{
  "author1": {
    "stance_annotations": [
      {
        "id": "[segment_id]",
        "label": "[chosen_label]",
        "reason": "[Justification in <50 words]"
      },
      ...
    ],
    "dogmatism_label": "[chosen_dogmatism_label]",
    "dogmatism_reason": "[Justification in <50 words]"
  },
  "author2": {
    "stance_annotations": [
      {
        "id": "[segment_id]",
        "label": "[chosen_label]",
```

```
      "reason": "[Justification in <50 words]"
    },
    ...
  ],
  "dogmatism_label": "[chosen_dogmatism_label]",
  "dogmatism_reason": "[Justification in <50 words]"
  }
}
'''
### Instructions for Effective Annotation

1. **Labeling Stance**: For each segment (including the original
   Reddit submission, comments, or replies) where "author1" or
   "author2" is mentioned, assign a stance label that best
   represents the stance expressed towards the discussed topic in
   the submission. This comprehensive approach ensures no relevant
   contribution by "author1" or "author2" is overlooked. Evaluate
   the stance based on the content's tone, argumentation, and
   engagement level with the topic.
2. **Providing Justification**: For each label assigned, include a
   concise reason, aiming for less than 50 words. Focus on the
   stance and argumentative indicators present in the text.
3. **Dogmatism Assessment**: After reviewing all segments from
   "author1" and "author2", assign a single dogmatism label
   reflecting the overall tone and approach in their contributions.
"""
```

## D   PROMPTS FOR FINETUNING SLMS

Fig. 7 and 8 shows the prompts used for finetuning SLMs for the stance and dogmatism classification tasks respectively.

---

**Stance Classification**

Analyze the stance of the post enclosed in square brackets.
Categorize each post into one of the following categories based on its stance:

- Somewhat In Favor
- Somewhat Against
- Stance Not Inferrable
- Strongly In Favor
- Strongly Against

and return the answer as one of the corresponding stance labels.

`[{data_point["stance_id_comment"]}]`

---

Figure 7: Prompt for stance classification, for finetuning SLMs.

> **User Dogmatism Identification**
>
> Analyze the comments of a user in conversation enclosed in square brackets.
> Categorize the opinion fluctuation of the user into one of the following categories based on its change:
>
> - Open to Dialogue
> - Firm but Open
> - Deeply Rooted
> - Flexible
>
> Return the answer as one of the corresponding dogmatism labels.
>
> `[{data_point["comments_string_for_dogmatism"]}]`

Figure 8: Prompt for dogmatism classification, for finetuning SLMs.

## E    SAMPLE OF USER INPUT PROMPT

```
"""
Now complete the given task for the respective authors i.e., author1
    respective ids are ['dhoxyz', 'f3pghji', 'f3tywb4', 'f3uomn2'].
    author2 respective ids are ['f3rt0bf', 'f3rqu2u'] for the data in
    json format
{
    "id":"dhoxyz",
    "title":"This sub should encourage anti vs. pro-gun discussions
        instead of shutting them down instantly",
    "content":"Honesly, I followed this sub especifically to take part in
        these discussions, but everytime I see a comment that even
        remotely suggests anti gun ideals or a discussion on the subject
        just gets ignored and downvoted to hell. Kind of expecting this to
         go the same way (my karma anus is ready, downvotes) , but I have
        to hope for healthy discussions on the subject.",
    "comments":[
        {
            "id":"f3p9n2c",
            "body":"I think the problem now is the two sides are at an
                impasse. Everytime there is a "compromise" pro gun loses
                something. Now days pro gun is interpreting the Constitution
                 more literal, which leaves even the most mild policies of
                anti gun as infringements. To further compound this anti gun
                 is only considering the most extreme measures. "Assault
                Weapons" bans, mandatory buybacks, red flag laws, etc.. I
                think at this point there is just nothing left to talk about
                . The middle ground is gone.",
            "replies":[
                {
                    "id":"f3pati9",
                    "replies":[
                        {
                            "id":"f3pdu44",
                            "body":"You are exactly right. I'm done with the
                                idea that there can be real compromise. We
                                should have at least gotten national reciprocity
                                 and shall-issue in every state in exchange for
                                what we've given up. Now you have to be a
                                goddamn lawyer to exercise your rights without
                                violating the law."
                        },
                        {
                            "id":"f3rt0bf",
```

```
                          "body":"I am prepared for UBCs, if they do this:
                          1. Lower the age to buy handguns to 18, nationwide.
                          2. Repeal the Hughes Amendment:
                          3. A FOPA-like ban on assault weapon bans (what the
                              FOPA did with a registry)
                          4. The punishment for violation is a monetary fine
                              only
                          5. A repeal of the GCA ban on foreign NFA weapons
                          6. A repeal of the National Minimum Drinking Age Act
                              of 1984"
                      }
                  ]
              },
              {
                  "id":"f3pd55z",
                  "body":"Everytime there is a "compromise" pro gun loses
                      something. That and today's compromise is tomorrow's
                      loophole to be closed. All such compromises do is push
                       that policy off until the next round."
              }
          ]
      },
      {
          "id":"f3paf0j",
          "body":"Yeah this sub it's not conducive to conversion. Its
              quickly devolving to little more than "Boogaloo" memes and
              shouting "SHALL. NOT." at each other. However,  as far as I
              know, the mods won't delete your thread and ban you from the
               sub for trying to have a good faith discussion,  like some
              of the gun control subs will.",
          "replies":[
              {
                  "id":"f3pusbm",
                  "body":"Unfortunately this sub's mod team takes a very
                      passive approach to moderation. With very little
                      effort they could make this sub into a  quality progun
                       meeting ground *without having to resort to
                      censorship*. Instead they promote low-effort memes and
                       endless duplication of posts through their inaction.
                      whubbard has the chops to resurrect this sub. Let's
                      see if he's up to the challenge.",
                  "replies":[
                      {
                          "id":"f3q8xj6",
                          "body":"We voted to ban memes last week. All about
                              rolling it out now.",
                          "replies":[
                              {
                                  "id":"f3qn4p8",
                                  "body":"Damn I might have to eat some crow
                                      here then..."
                              }
                          ]
                      }
                  ]
              }
          ]
      },
      {
          "id":"f3pafqa",
          "body":"Found the gun grabber!!",
          "replies":[
              {
                  "id":"f3pcw4h",
                  "body":"Witch hunter."
```

```
1080                       }
1081                   ]
1082           },
1083           {
1084               "id":"f3pal5l",
1085               "body":"I see people have discussions when it makes sense to.
1086                   Not much reason to spend time responding to the same gun
1087                   control measures over and over though."
1088           },
1089           {
1090               "id":"f3paw3h",
1091               "body":"I get where you're coming from, but people's ability to
1092                   protect themselves and own their own property isn't
1093                   something that is compromisable. Anything less, and they
1094                   cease to own their own property. It's like breathing, there
1095                   can be nothing less than total ability to breath when and
1096                   how someone wants. It's just that simple."
1097           },
1098           {
1099               "id":"f3pax9m",
1100               "body":"My take on this, What kind of open discussion is
1101                   possible for a right that is guaranteed and most importantly
1102                   , not to be infringed upon? They're making all these
1103                   unlawful laws to portray it as it's somehow legitimate. They
1104                    are not, We are at an apex, to which both political
1105                   spectrums and even us to a degree are liable for.\nI
1106                   certainly believe both sides are waiting for this to boil
1107                   over so each can finger point. I just speculate it's going
1108                   to be the hell humanity been whispering about but never
1109                   thought it would ever occur."
1110           },
1111           {
1112               "id":"f3pb6ny",
1113               "body":"The time for discussion is over."
1114           },
1115           {
1116               "id":"f3pfqwq",
1117               "body":"I don't know what you're talking about. Sure people
1118                   downvote, but they also talk. We get "why do you need guns"
1119                   posts at least weekly, and several people will engage in
1120                   actual conversation with them, citing facts, clearing up
1121                   statistics, and telling stories to illustrate why this is
1122                   important to them, but they are usually met with "you stupid
1123                    @#$%, you think you're Rambo" or something equally clever.
1124                   People who come here to discuss and learn will be treated
1125                   well. People who are just trolling are treated like trolls
1126                   .",
1127               "replies":[
1128                   {
1129                       "id":"f3pghji",
1130                       "body":"I made this post because I'm always seeing
1131                           rational, conversation seeking comments getting blown
1132                           to downvote hell.",
1133                       "replies":[
                             {
                           "id":"f3pi9xv",
                           "body":"[Like this one?](https://www.reddit.com/r/progun
                               /comments/dhcu92/yup/f3p75tg/)> One smart man in a
                               sub full of... welp... "strong opinions". You start
                               off with arrogance, as the sole arbiter of what
                               constitutes a "smart man". Then you back it up with
                               a dismissive swipe at what you term "strong opinions
                               ".> Every other country can see that PROPER gun
                               control reduces gun violence by a ton, More
                               arrogance. False equivalence. Unsupported claims.>
```

```
                            but the US refuses to let go of it's antique laws In
                             a shocking turn of events, more arrogance.> Fully
                            aware that this is a fully pro gun sub, willing to
                            take the downvotes in order to spark a discussion
                            and crack some heads. You aren't the first arrogant
                            asshole to grace this sub with posts like this. Try
                            bringing something other than your own self-
                            importance to the discussion. Edit: And then there's
                             [this gem](https://www.reddit.com/r/
                            unpopularopinion/comments/d3w5z1/
                            people_living_in_the_us_are_living_in_one_of_the/
                            f06r3sg/.> Wanna feel like you could be shot at
                            every single moment? Move to the US, it'll prob
                            happen to you either as a bystander, or you'd be
                            shot by a random citizen (sometimes police)."
                    },
                    {
                        "id":"f3pj8k0",
                        "body":"As is tradition. We're done with that
                            condescending bullshit from antis, you dont come
                             here for good faith discussion and whether you
                            get a reasonable response or not, nothing ever
                            changes, easier to downvote you and move on
                            because we get the same treatment anytime we
                            attempt to speak out in anti subs."
                    },
                    {
                        "id":"f3plgf4",
                        "body":"If downvotes hurt your feelings, you shouldn
                            't be on reddit. People tend to downvote
                            anything they disagree with (which is why some
                            subs specifically ask you to only downvote
                            things that contribute nothing to the discussion
                            ). It's a bad habit, but that's the way it is.
                            People downvote and *still* enage. You want to
                            post a view contrary to the prevailing view of
                            the sub, take your lumps and participate in what
                             conversation you are offered. But if you're
                            only here to preach about how stupid, misguided,
                             unevolved, uneducated, irrational, and/or
                            violent we are, don't expect a polite response."
                    },
                    {
                        "id":"f3tcgf1",
                        "body":"An arrogant Israeli trying to tell another
                            nation how they should be run. You're just a
                            walking stereotype aren't you? And before you
                            say anything, I popped into your comment history
                            . That's where the calling you Israeli comes
                            from.",
                        "replies":[
                            {
                                "id":"f3tywb4",
                                "body":"I thought that trying to tell other
                                    nations how they should run was your guys'
                                    s stereotype.",
                                "replies":[
                                    {
                                        "id":"f3u0vkq",
                                        "body":"No we go in and try to make them
                                            work our way."
                                    }
                                ]
                            }
                        ]
```

```
                        }
                    ]
                }
            ]
        },
        {
            "id":"f3pzseh",
            "body":"It's a little unfortunate but the grabbers who come on
                here tend to be intellectually dishonest and/or uninformed.
                There was some Australian post a few days ago that pretty
                much asked why we like our guns more than children. No
                discussion to be had there. There's also some posts that
                clearly demonstrate the poster should inform himself or
                herself a little."
        },
        {
            "id":"f3rqu2u",
            "body":"Actually, do that. It shows everyone that they tend to
                be crazy, unstable, ignorant, stereotyping, arrogant
                bastards who hate black people with a hair trigger."
        },
        {
            "id":"f3t7tgg",
            "body":"Welcome to reddit, home of every single safe place for
                anything that doesnt violate the TOS. At least its slightly
                better than r/politics"
        },
        {
            "id":"f3unt9z",
            "body":"This isn't r/gundebate. This is a pro gun subreddit.
                That said, we do allow some debate provided it remains civil
                .",
            "replies":[
                {
                    "id":"f3uomn2",
                    "body":"Sadly tho, r/gundebate is pretty dead..."
                }
            ]
        },
        {
            "id":"f4dip6o",
            "body":"Anything else you want to give away for free?"
        }
    ]
}
```

## F  SITUATIONS LEADING TO LLM ANNOTATION ERRORS AND INCONSISTENCIES

Before proceeding with LLM annotation using larger models, we first tested other versions of GPT and Mistral models, such as GPT-3.5 and Mistral-small and medium. However, we found that these models failed to produce annotations in the desired format. Below are some specific situations where LLMs were prone to errors:

- **System Prompt Clarity:** The importance of a clear and precise system prompt cannot be overstated. When the prompt lacked clarity, LLMs often generated annotations for unspecified authors, indicating confusion about the task requirements.

- **Understanding Conversation Structure:** Without providing a clear example of the conversation structure, none of the LLMs were able to understand the task properly. This demonstrates the need for explicit guidance when dealing with complex conversation data.

- **Interface Issues:** Using an interface to facilitate LLM annotation proved problematic. After processing 2 to 3 examples, LLMs began providing annotations for previous user IDs, even when presented with new conversations. This suggests that the model lost track of the task and context.

- **Consistency in Annotations:** For smaller conversations, different LLMs tended to produce similar annotations. However, as the conversations grew longer, the annotations became inconsistent across different models, indicating challenges in maintaining accuracy over extended discourse.

- **Confusion with Author IDs:** Occasionally, LLMs confuse author IDs, resulting in missed stance labels for certain authors (as shown in Fig. 3 (left) in the main paper). Additionally, there were minor errors in key naming (e.g., 'label' vs. 'body' as shown in Fig. 3 (right) in the main paper), which further highlighted the model's limitations.

# G  SAMPLES OF JSON OUTPUTS FROM LLMs

## G.1  SAMPLE JSON OUTPUT FROM LLM: GPT-4 ZERO-SHOT

```
{
    "author1": {
    "stance_annotations": [
    {
        "id": "dhoxyz",
        "label": "somewhat_in_favor",
        "reason": "Advocates for open discussions, showing interest in
            dialogue."
    },
    {
        "id": "f3pghji",
        "label": "somewhat_in_favor",
        "reason": "Expresses desire for rational discussions, despite
            downvotes."
    },
    {
        "id": "f3tywb4",
        "label": "somewhat_against",
        "reason": "Sarcastic retort, but lacks strong emotional language
            ."
    },
    {
        "id": "f3uomn2",
        "label": "stance_not_inferrable",
        "reason": "Comment is factual, no clear stance on the debate
            itself."
    }
    ],
    "dogmatism_label": "Open to Dialogue",
    "dogmatism_reason": "Shows interest in discussion, acknowledges
        downvotes."
    },
    "author2": {
    "stance_annotations": [
    {
        "id": "f3rt0bf",
        "label": "somewhat_in_favor",
        "reason": "Proposes conditions for UBCs, indicating some openness
            ."
    },
    {
        "id": "f3rqu2u",
        "label": "strongly_against",
```

```
1296          "reason": "Uses strong negative descriptors for opponents,
1297              indicating firm stance."
1298      }
1299      ],
1300      "dogmatism_label": "Firm but Open",
1301      "dogmatism_reason": "Mixed; proposes solutions but also dismisses
1302          opposition."
1303  }
```

## G.2  SAMPLE JSON OUTPUT FROM LLM: GPT-4 ONE-SHOT

```
1308  {
1309      "author1": {
1310      "stance_annotations": [
1311      {
1312          "id": "dhoxyz",
1313          "label": "somewhat_in_favor",
1314          "reason": "Advocates for open discussions, indicating a pro-
                  discussion stance."
1315      },
1316      {
1317          "id": "f3pghji",
1318          "label": "somewhat_in_favor",
1319          "reason": "Expresses desire for rational conversation, showing a
                  supportive stance for dialogue."
1320      },
1321      {
1322          "id": "f3tywb4",
1323          "label": "somewhat_against",
1324          "reason": "Uses sarcasm to counter a stereotype, indicating a
                  stance against the opposing view."
1325      },
1326      {
1327          "id": "f3uomn2",
1328          "label": "somewhat_in_favor",
1329          "reason": "Expresses disappointment in lack of active debate,
                  supporting open discussion."
1330      }
1331      ],
1332      "dogmatism_label": "Open to Dialogue",
1333      "dogmatism_reason": "Shows a consistent preference for discussion and
              engagement, without dogmatic assertions."
1334      },
1335      "author2": {
1336      "stance_annotations": [
1337      {
1338          "id": "f3rt0bf",
1339          "label": "somewhat_in_favor",
1340          "reason": "Lists conditions for agreement, showing openness to
                  compromise."
1341      },
1342      {
1343          "id": "f3rqu2u",
1344          "label": "strongly_against",
1345          "reason": "Uses strong language to describe opponents, indicating
                  a firm stance against them."
1346      }
1347      ],
1348          "dogmatism_label": "Firm but Open",
1349          "dogmatism_reason": "While showing strong opinions, also lists
                  specific conditions for compromise."
       }
```

```
1350
1351     }
1352
```

### G.3 SAMPLE JSON OUTPUT FROM LLM: GPT-4 LARGE FEW-SHOT

```
1355   {
1356       "author1": {
1357       "stance_annotations": [
1358       {
1359           "id": "dhoxyz",
1360           "label": "somewhat_in_favor",
1361           "reason": "Expresses desire for open discussions, indicating a
1362               stance in favor of dialogue."
1363       },
1364       {
1365           "id": "f3pghji",
1366           "label": "somewhat_in_favor",
1367           "reason": "Advocates for rational conversation despite observing
1368               negative reactions."
1369       },
1370       {
1371           "id": "f3tywb4",
1372           "label": "somewhat_against",
1373           "reason": "Uses sarcasm to counter a stereotype, showing a stance
1374               against it."
1375       },
1376       {
1377           "id": "f3uomn2",
1378           "label": "stance_not_inferrable",
1379           "reason": "Comment on the state of another subreddit does not
1380               reveal a stance on the gun debate."
1381       }
1382       ],
1383       "dogmatism_label":  "Open to Dialogue",
1384       "dogmatism_reason": "Shows a consistent
1385       preference for discussion over confrontation."
1386       },
1387       "author2": {
1388       "stance_annotations": [
1389       {
1390           "id": 'f3rt0bf',
1391           "labe"': 'somewhat_in_favor",
1392           "reason": 'Lists conditions for accepting UBCs, indicating some
1393               openness to compromise."
1394       },
1395       {
1396           "id": 'f3rqu2u',
1397           "label": "strongly_against",
1398           "reason": "Uses derogatory language to describe opponents,
1399               showing strong opposition."
1400       }
1401       ],
1402       "dogmatism_label": "Firm but Open",
1403       "dogmatism_reason": "While showing strong opinions, also suggests
           conditions for policy acceptance."
       }
   }
```

### G.4 SAMPLE JSON OUTPUT FROM LLM: MISTRAL LARGE ZERO-SHOT

```
   {
       "author1": {
       "stance_annotations": [
```

```
     {
          "id": "dhoxyz",
          "label": "somewhat_in_favor",
          "reason": "Author expresses desire for open discussions on anti-
              gun ideals."
     },
     {
          "id": "f3pghji",
          "label": "stance_not_inferrable",
          "reason": "Author is commenting on downvotes, not expressing
              stance."
     },
     {
          "id": "f3tywb4",
          "label": "stance_not_inferrable",
          "reason": "Author is making a stereotypical comment, not
              expressing stance."
     },
     {
          "id": "f3uomn2",
          "label": "stance_not_inferrable",
          "reason": "Author is commenting on the state of another subreddit
              , not expressing stance."
     }
     ],
     "dogmatism_label": "Open to Dialogue",
     "dogmatism_reason": "Author expresses desire for open discussions and
          does not dismiss opposing views."
     },
     "author2": {
     "stance_annotations": [
     {
          "id": "f3rt0bf",
          "label": "strongly_in_favor",
          "reason": "Author lists conditions for accepting UBCs, showing
              strong pro-gun stance."
     },
     {
          "id": "f3rqu2u",
          "label": "strongly_in_favor",
          "reason": "Author"s comment is dismissive of anti-gun advocates,
              showing strong pro-gun stance."
     }
     ],
     "dogmatism_label": "Deeply Rooted",
     "dogmatism_reason": "Author"s comments are dismissive of opposing
          views and show a strong, unchangeable belief in pro-gun ideals."
     }
}
```

## G.5 SAMPLE JSON OUTPUT FROM LLM: MISTRAL LARGE ONE-SHOT

```
{
     "author1": {
     "stance_annotations": [
     {
          "id": "dhoxyz",
          "label": "somewhat_in_favor",
          "reason": "Expresses desire for open discussions on gun control,
              but follows pro-gun subreddit."
     },
     {
          "id": "f3pghji",
```

```
        "label": "stance_not_inferrable",
        "reason": "Comments on downvotes, but no clear stance on gun
            control."
    },
    {
        "id": "f3tywb4",
        "label": "stance_not_inferrable",
        "reason": "Comment about stereotypes, but no clear stance on gun
            control."
    },
    {
        "id": "f3uomn2",
        "label": "stance_not_inferrable",
        "reason": "Mentions r/gundebate being dead, but no clear stance
            on gun control."
    }
    ],
    "dogmatism_label": "Open to Dialogue",
    "dogmatism_reason": "Seeks open discussions and does not express
        strong dogmatic views."
    },
    "author2": {
    "stance_annotations": [
    {
        "id": "f3rt0bf",
        "label": "somewhat_in_favor",
        "reason": "Lists conditions for supporting UBCs, showing openness
            to compromise."
    },
    {
        "id": "f3rqu2u",
        "label": "strongly_in_favor",
        "reason": "Strong negative language against gun control advocates
            , suggesting strong pro-gun stance."
    }
    ],
    "dogmatism_label": "Firm but Open",
    "dogmatism_reason": "Expresses strong pro-gun views but open to
        compromise under certain conditions."
    }
}
```

## G.6 SAMPLE JSON OUTPUT FROM LLM: MISTRAL LARGE FEW-SHOT

```
{
    "author1": {
    "stance_annotations": [
    {
        "id': "dhoxyz",
        "label": "somewhat_in_favor",
        "reason": "Expresses desire for open discussions on anti vs. pro-
            gun debates."
    },
    {
        "id": "f3pghji",
        "label": "stance_not_inferrable",
        "reason": "Comments on downvotes, but no clear stance on the
            topic."
    },
    {
        "id": "f3tywb4",
        "label": "stance_not_inferrable",
        "reason": "Makes a stereotypical comment, but no clear stance on
            the topic."
```

```
      },
    {
        "id": "f3uomn2",
        "label": "stance_not_inferrable",
        "reason": "Mentions r/gundebate being dead, but no clear stance
            on the topic."
    }
    ],
    "dogmatism_label": "Open to Dialogue",
    "dogmatism_reason": "Seeks open discussions and engages in
        conversation without strong dogmatic language."
    },
    "author2": {
    "stance_annotations": [
    {
        "id": "f3rt0bf",
        "label": "somewhat_in_favor",
        "reason": "Lists conditions for supporting UBCs, showing openness
            to discussion."
    },
    {
        "id": "f3rqu2u",
        "label": "strongly_against",
        "reason": "Uses derogatory language to express strong opposition
            to 'grabbers'."
    }
    ],
    "dogmatism_label": "Firm but Open",
    "dogmatism_reason": "Expresses strong opinions but also shows
        willingness to consider certain conditions for compromise."
    }
}
```

# H  DETAILS OF SMALL LANGUAGE MODELS AND HYPER-PARAMETER SETTINGS

**LLaMA** models (Touvron et al., 2023a) are decoder-only LLMs trained on 1.6 trillion tokens from a mixture of corpora including C4, English CommonCrawl, Wikipedia, Github, and more. We use two versions of models in our study: LLaMa-2-7B (Touvron et al., 2023b) and LLaMa-3-8B and their instruction-tuned variants.

**Falcon** models (Almazrouei et al., 2023) are decoder-only LLMs trained on $\geq 1$ trillion tokens of text, particularly emphasizing the RefinedWeb corpus. For Falcon, we use the pretrained and instruction-tuned 7B parameter variants in our study.

**Vicuna** model (Chiang et al., 2023) is finetuned from the LLaMA 7B model on approximately 70K user-shared conversations gathered from ShareGPT.com and we used the 7B parameter variants.

**Implementation details for reproducibility.** All experiments were conducted on a machine equipped with an NVIDIA A100 GPU with 80 GB of GPU RAM, partitioned into two devices of 40 GB each. We employed 4-bit quantization with normalized floating precision (nf4) from the bitsandbytes library [3]. Additionally, we utilized LoRA (Hu et al., 2021) with a rank of 64 and an alpha value of 16 during task-based instruction-tuning. Finally, we use PEFT (Parameter Efficient Finetuning) [4] library to train LLMs with the SFTT (Supervised Finetuning Trainer) setting. To further enhance performance, we divided the training dataset into a validation set comprising a randomly chosen 10% subset from the training set, used exclusively for hyperparameter tuning.

---

[3] https://pypi.org/project/bitsandbytes/
[4] https://github.com/huggingface/peft

# I BASELINE (UN-FINE-TUNED) MODEL PERFORMANCE

**Stance Detection**

Table 4: Classification Report for GPT-4 Few-shot as target labels: Un-finetuned performance: weighted F1 score for Stance classification using SLMs on USDC test set.

| Class | Precision | Recall | F1-Score | Support |
|---|---|---|---|---|
| Somewhat Against | 0.26 | 0.67 | 0.38 | 400 |
| Somewhat In Favor | 0.45 | 0.21 | 0.28 | 624 |
| Stance Not Inferrable | 0.35 | 0.11 | 0.16 | 454 |
| Strongly Against | 0.25 | 0.38 | 0.30 | 261 |
| Strongly In Favor | 0.13 | 0.02 | 0.03 | 128 |
| **Accuracy** | | 0.29 | | 1867 |
| **Macro avg** | 0.29 | 0.28 | 0.23 | 1867 |
| **Weighted avg** | 0.33 | 0.29 | 0.26 | 1867 |

Table 5: Classification Report for Mistral Large few-shot as target labels: Un-finetuned performance: weighted F1 score for Stance classification using SLMs on USDC test set.

| Class | Precision | Recall | F1-Score | Support |
|---|---|---|---|---|
| Somewhat Against | 0.20 | 0.69 | 0.31 | 316 |
| Somewhat In Favor | 0.39 | 0.24 | 0.30 | 458 |
| Stance Not Inferrable | 0.41 | 0.08 | 0.14 | 567 |
| Strongly Against | 0.29 | 0.32 | 0.30 | 336 |
| Strongly In Favor | 0.31 | 0.02 | 0.04 | 190 |
| **Accuracy** | | 0.26 | | 1867 |
| **Macro avg** | 0.32 | 0.27 | 0.22 | 1867 |
| **Weighted avg** | 0.34 | 0.26 | 0.23 | 1867 |

Table 6: Classification Report for Majority Voting as target labels: Un-finetuned performance: weighted F1 score for Stance classification using SLMs on USDC test set.

| Class | Precision | Recall | F1-Score | Support |
|---|---|---|---|---|
| Somewhat Against | 0.30 | 0.71 | 0.42 | 443 |
| Somewhat In Favor | 0.41 | 0.20 | 0.27 | 625 |
| Stance Not Inferrable | 0.34 | 0.09 | 0.14 | 452 |
| Strongly Against | 0.26 | 0.39 | 0.31 | 256 |
| Strongly In Favor | 0.19 | 0.03 | 0.06 | 91 |
| **Accuracy** | | 0.31 | | 1867 |
| **Macro avg** | 0.30 | 0.28 | 0.24 | 1867 |
| **Weighted avg** | 0.34 | 0.31 | 0.27 | 1867 |

**Dogmatism Identification**

Table 7: Classification Report for Majority Voting as target labels: Un-finetuned performance: weighted F1 score for Dogmatism classification using SLMs on USDC test set.

| Class | Precision | Recall | F1-Score | Support |
|---|---|---|---|---|
| Deeply Rooted | 0.17 | 0.54 | 0.26 | 28 |
| Firm but Open | 0.50 | 0.25 | 0.34 | 131 |
| Flexible | 0.00 | 0.00 | 0.00 | 14 |
| Open to Dialogue | 0.48 | 0.55 | 0.51 | 134 |
| **Accuracy** | | 0.40 | | 307 |
| **Macro avg** | 0.29 | 0.33 | 0.28 | 307 |
| **Weighted avg** | 0.44 | 0.40 | 0.39 | 307 |

## J  SLM FINETUNING: AUC (AREA UNDER THE CURVE) ANALYSIS

Fig. 9 illustrates the confusion matrix for dogmatism detection for LLaMa-3-8B finetuning and instruction-tuning. We make the following observations from Fig. 9: 1) For both finetuning and instruction-tuning, there are significant misclassifications, especially for the "Deeply Rooted" and "Flexible" labels, with both having zero accuracy and F1-scores. While "Firm but Open" and "Open to Dialogue" perform moderately better, with accuracies of 48.7% and 64.4% respectively. The confusion matrix indicates substantial confusion to distinguish between intermediate levels of dogmatism, such as "Firm but Open" and "Open to Dialogue". We further report the ROC curve shows the trade-off between the true positive rate (TPR) and false positive rate (FPR) for each class for stance and dogmatism tasks, in Figs. 10 and. 11. The area under the ROC curve (AUC) measures the model's ability to distinguish between classes.

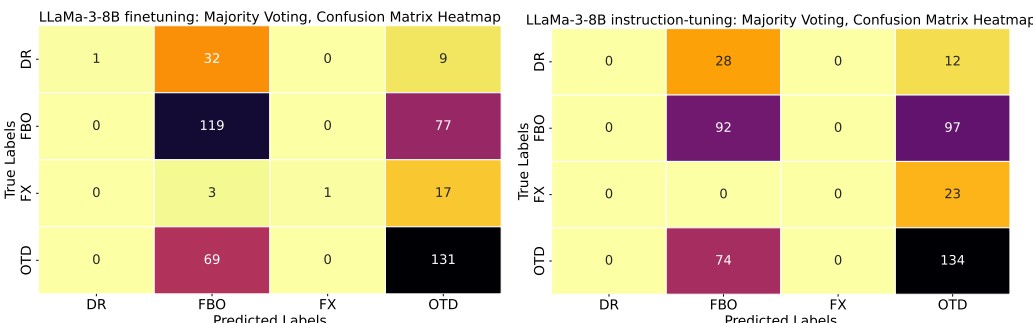

Figure 9: Confusion matrix for LLaMa-3-8B Dogmatism detection models on USDC test set: finetuning (left) and instruction-tuning (right). Here, DR: Deeply Rooted, FX: Flexible, FBO: Firm but Open, OTD: Open to Dialogue

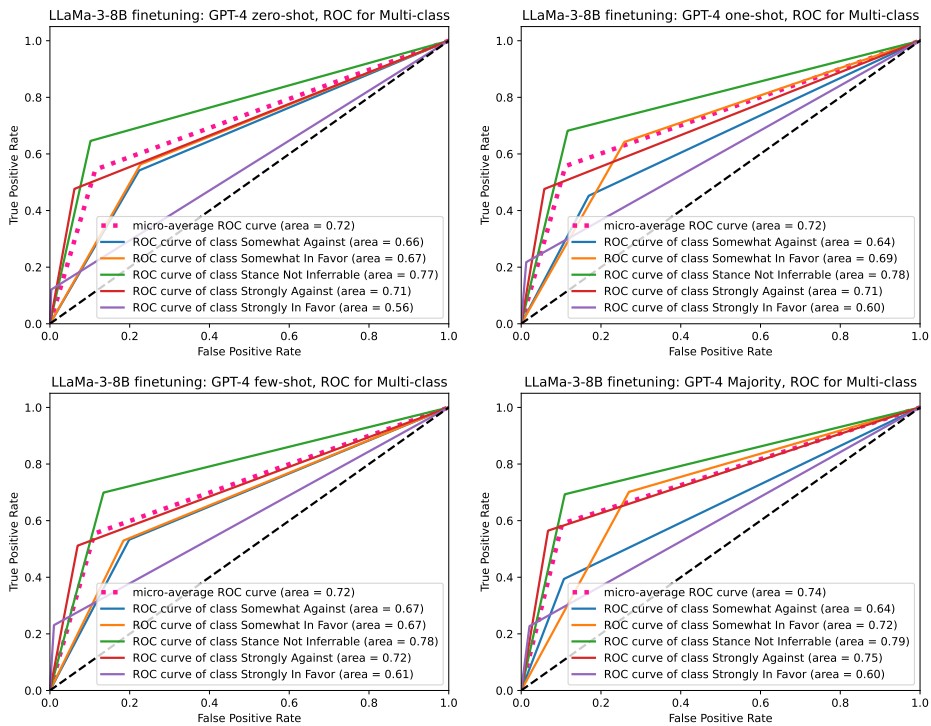

Figure 10: LLaMa-3-8B finetuning for stance detection task: Visualize the ROC curves for each class along with their AUC values for GPT-4 annotations across zero-shot, one-shot, few-shot and majority labels.

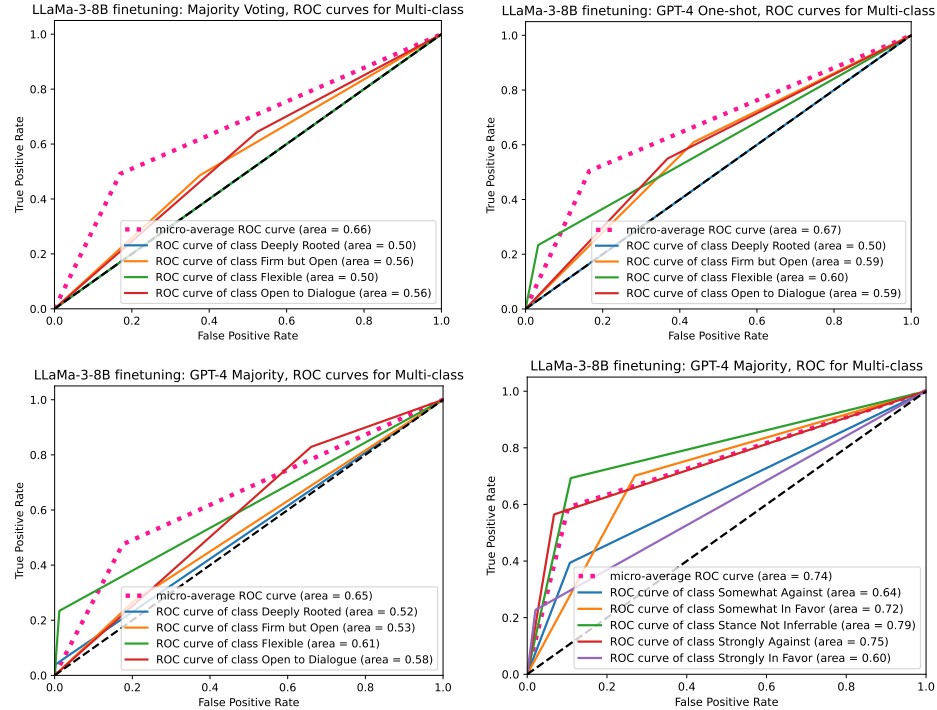

Figure 11: LLaMa-3-8B finetuning for dogmatism task: Visualize the ROC curves for each class along with their AUC values for GPT-4 annotations across zero-shot, one-shot, few-shot and majority labels.

## K   SLM INSTRUCTION-TUNING: AUC (AREA UNDER THE CURVE) ANALYSIS

Fig. 12 shows the ROC curve trade-off between the true positive rate (TPR) and false positive rate (FPR) for each class for stance task using LLaMa-3-8B instruction-tuning. This instruction-tuning is performed on GPT-4 (zero-shot, one-shot, few-shot) and majority voting labels from the USDC dataset. We make the following observations from Fig. 12: 1) Across all four settings, the area under the curve (AUC) for all stance labels is $>= 0.5$. This indicates that the model predicts each stance label more accurately than random guessing for all classes. 2) Among all settings, the majority voting labels from the USDC dataset show a higher AUC for each class compared to zero-shot, one-shot, and few-shot labels. 3) Among all stance classes, the "Stance Not Inferrable" class has the highest AUC (0.8), while the "Strongly In Favor" class has the lowest AUC (0.6). Overall, LLaMa-3-8B instruction-tuning demonstrates superior performance in the stance detection task. However, there is still significant room for improvement in understanding user opinions from text segments.

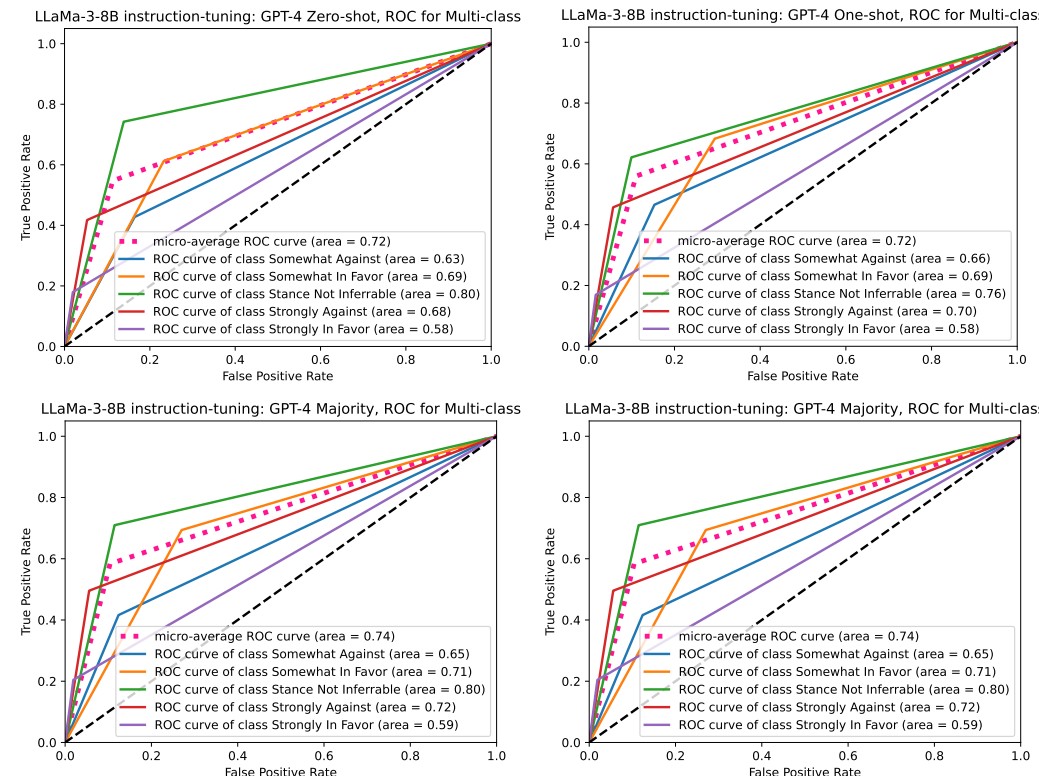

Figure 12: LLaMa-3-8B instruction-tuning for stance detection task: Visualize the ROC curves for each class along with their AUC values for GPT-4 annotations across zero-shot, one-shot, few-shot and majority labels.

## L  LOST IN THE MIDDLE

To analyze the "lost in the middle" Liu et al. (2024) phenomenon in our LLM-based user-stance annotations, for a given user, we divided the data into time segments and calculated inter-annotator agreement (IAA) using Cohen's Kappa scores across different models and settings. The data was segmented based on the submission_id, author_id, and stance_id_timestamp. For each group (i.e., each combination of submission_id and author_id), the timestamps were divided into equal segments. The number of entries for each group was divided by the desired number of segments (3), and the division was done as evenly as possible, with each segment containing a roughly equal number of time-stamped entries. Fig. 13 in Appendix reports the comparison statistics of IAA scores for the stance detection task across initial, middle, and later time stamps. From Fig. 13, we observe that the analysis across different time segments, especially when divided into three segments, clearly demonstrates that the "lost in the middle" phenomenon is marginal.

The partial decrease in inter-annotator agreement during the middle parts of the conversations suggests that as conversations progress, models might face challenges in maintaining consistent agreement; however, the decrease in agreement scores is minimal. The recovery in agreement towards the final segments could indicate that as conversations start to conclude, they become more focused, or that the models are better able to align on concluding statements. This trend underscores the importance of considering segment-based analysis when evaluating model performance over long-form conversations. When comparing the model-generated annotations with human annotations, it becomes evident that we do not encounter the "lost in the middle" problem. The human annotations demonstrate a consistent level of inter-annotator agreement (IAA) across all three segments—initial, middle, and final. This suggests that human annotators maintain a steady understanding and agreement throughout the conversation, regardless of its length or complexity.

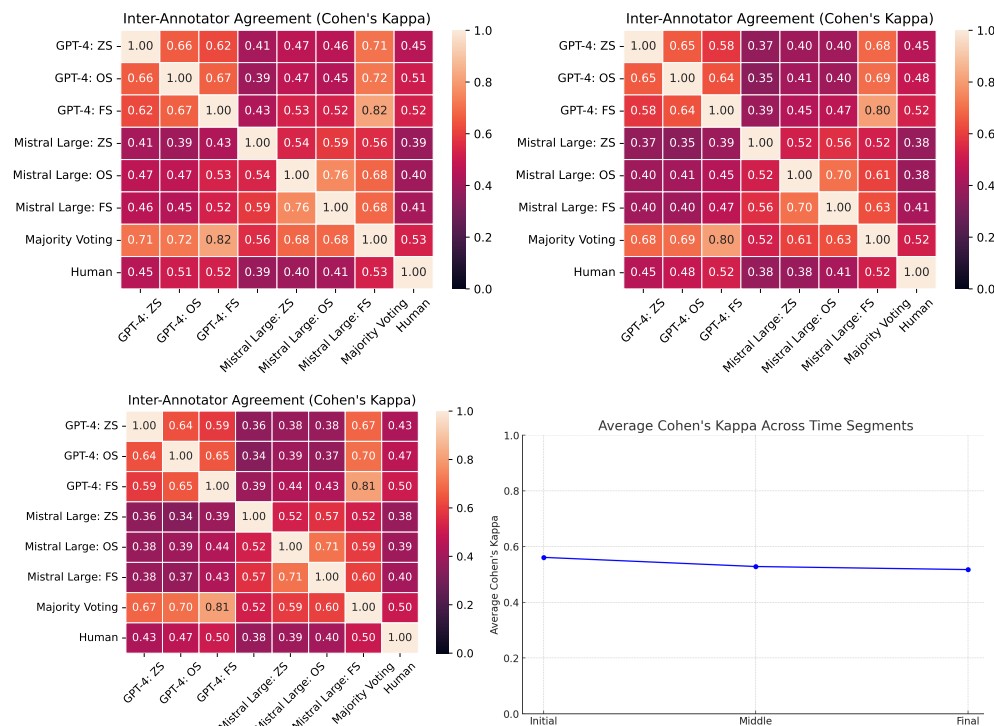

Figure 13: The inter-annotator agreement (IAA) on the USDC test dataset was measured using Cohen's Kappa score across three segments: initial, middle, and later time stamps. The top two rows represent the initial and middle time stamps, while the bottom left corresponds to the later time stamp. The bottom right reports the average Kappa score across all time segments.

## M RECENCY BIAS

Fig. 14 reports IAA scores, which contains a matrix of Cohen's Kappa scores across different models and settings, including GPT-4 Few-Shot (FS), Mistral Large FS, Majority Voting, as well as GPT-4 FS PC and Mistral Large FS PC (here, PC denotes prior context). From the figure, we observe that the agreement between GPT-4 FS and Majority Voting is higher when the full conversation is considered (0.75) compared to when only prior context is used. The agreement between GPT-4 FS PC and Mistral Large FS PC (both based on prior context) is lower than when using the full context, indicating that prior context alone may not capture all the necessary nuances for consistent annotation.

## N SLM FINETUNING: TRANSFER LEARNING PERFORMANCE

### N.1 STANCE DETECTION EVALUATION ON SPINOS DATASET:

To evaluate the quality of LLM generated annotations, we perform transfer learning by finetuning the SLMs on the USDC dataset. We then test the model's performance on the SPINOS dataset for a 5-class Stance detection task, as described by Sakketou et al. (2022). We use the USDC training dataset. For testing, we use the SPINOS dataset, which consists of 3,238 post level examples across five stance labels.

Fig. 15 in Appendix N illustrates the confusion matrix for stance detection for LLaMa-3-8B finetuning on USDC and testing on SPINOS. We make the following observations from Fig. 15: 1) There is a significant misclassification across all classes, with the "Stance Not Inferrable" label being the most commonly predicted class, resulting in many false positives for this label. 2) The model performs best in terms of accuracy for three stance classes: "Somewhat In Favor" (0.456), "Strongly Against" (0.400), and "Somewhat Against" (0.381), while performing the worst for the "Strongly In Favor" stance (0.115). These overlaps suggest challenges in distinguishing whether a post contains stance or

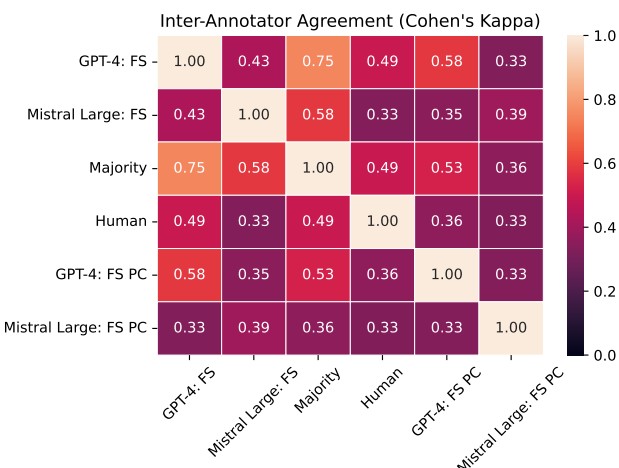

Figure 14: Inter-annotator agreement (IAA) on the test dataset was calculated for both the full conversations and the prior context for a given user. In this context, "GPT-4 FS PC" and "Mistral Large: FS PC" refer to the annotations based on prior context.

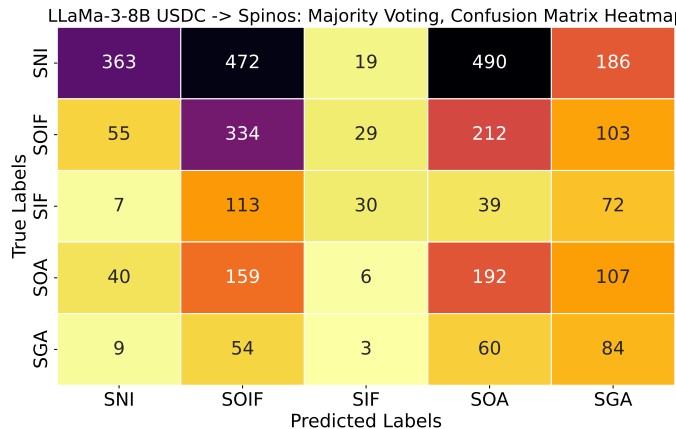

Figure 15: Confusion matrix for LLaMa-3-8B Stance detection models on SPINOS test set: finetuning on USDC and test it on SPINOS. SOA: Somewhat Against, SOIF: Somewhat In Favor, SNI: Stance Not Inferrable, SGA: Strongly Against, SIF: Strongly In Favor.

not, indicating a need for enhanced feature representation and clearer class definitions to improve model performance.

In comparison to the SPINOS dataset results reported in the paper by Sakketou et al. (2022), where the best model (traditional machine learning classifier) achieved an F1-score of 0.341, a random baseline achieved 0.230, and a majority baseline achieved 0.124. Our approach using LLaMa-3-8B finetuning on the USDC dataset achieved a weighted F1-score of 0.320 on SPINOS. This score is close to the best model performance on the SPINOS dataset, indicating that our LLM-generated annotations on the USDC dataset are close in quality to human annotations. It is important to note that our weighted F1-score is significantly impacted by the "Stance Not Inferrable" class, which comprises the majority of samples in the SPINOS dataset. Our finetuned SLM struggled to classify this class accurately, leading to a lower overall weighted F1-score.

We also validated the SPINOS performance using other SLMs such as LLaMa-3-8B-Instruct, LLaMa-2-7B, LLaMa-2-7B-Chat, and Vicuna-7B models. Figs. 16, 17, 18 and 19 in Appendix N display these model results. These figures indicate that these models report weighted F1-scores of 0.320, 0.305, 0.286, and 0.291 respectively. These results show that all models perform better than the random and majority baselines. Additionally, the LLaMa-3-8B-Instruct model's performance is close to the SPINOS benchmark on the 5-class stance detection task.

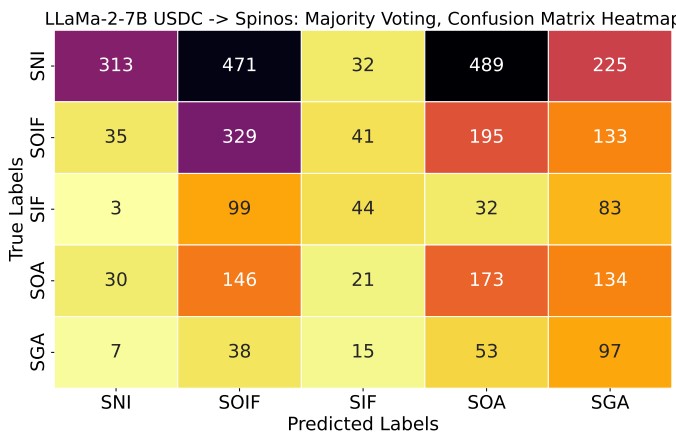

Figure 16: Confusion matrix for LLaMa-3-8B-instruct Stance detection models on SPINOS test set: finetuning on USDC and test it on SPINOS. SOA: Somewhat Against, SOIF: Somewhat In Favor, SNI: Stance Not Inferrable, SGA: Strongly Against, SIF: Strongly In Favor.

Figure 17: Confusion matrix for LLaMa-2-7B Stance detection models on SPINOS test set: finetuning on USDC and test it on SPINOS. SOA: Somewhat Against, SOIF: Somewhat In Favor, SNI: Stance Not Inferrable, SGA: Strongly Against, SIF: Strongly In Favor.

Fig. 15 illustrates the confusion matrix for Stance detection for LLaMa-3-8B finetuning on USDC and transfer learning on SPINOS. We also validated the SPINOS performance using other SLMs such as LLaMa-3-8B-Instruct, LLaMa-2-7B, LLaMa-2-7B-Chat, and Vicuna-7B models. Figs. 16, 17, 18 and 19 display these model results.

## N.2 SLM FINETUNING: TRANSFER LEARNING PERFORMANCE ON MT-CDS DATASET

The transfer learning accuracies using the USDC dataset on the MT-CSD dataset (Niu et al., 2024) is tailored for stance detection in multi-turn conversations with multiple targets, addressing different aspects of stance detection. This dataset consists of human annotated labels across 5 stance datasets (Biden, Bitcoin, SpaceX, Tesla, and Trump) in testing. This MT-CDS stance dataset contains 3-class labels such as favor, against and neutral. Therefore, we combined our Strongly Against and Somewhat Against as one class, Strongly In Favor and Somewhat In Favor as one class and Stance Not Inferrable as one class. Below are the accuracies we obtained on 5 datasets. From the Table 8, we observe that our transfer learning results are closer or performing better than results reported in Table 6 of Niu et al. (2024). This implies that our LLM generated annotations are closer to human-level performance on MT-CDS stance detection dataset.

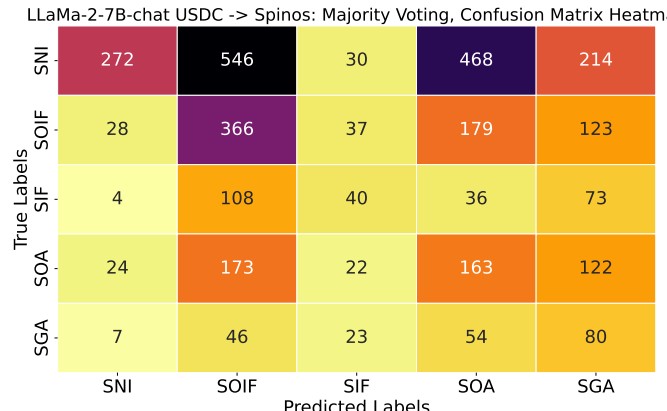

Figure 18: Confusion matrix for LLaMa-2-7B-chat Stance detection models on SPINOS test set: finetuning on USDC and test it on SPINOS. SOA: Somewhat Against, SOIF: Somewhat In Favor, SNI: Stance Not Inferrable, SGA: Strongly Against, SIF: Strongly In Favor.

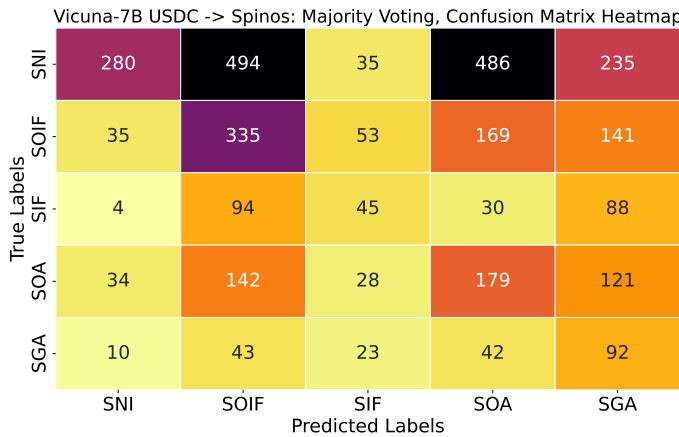

Figure 19: Confusion matrix for Vicuna-7B Stance detection models on SPINOS test set: finetuning on USDC and test it on SPINOS. SOA: Somewhat Against, SOIF: Somewhat In Favor, SNI: Stance Not Inferrable, SGA: Strongly Against, SIF: Strongly In Favor.

Table 8: Stance Detection Evaluation on MT-CDS Dataset: USDC dataset in training and MT-CDS dataset in testing.

| Dataset | Best Accuracy | USDC accuracy |
|---------|---------------|---------------|
| Biden | 45.09 | 46.60 |
| Bitcoin | 56.95 | 51.40 |
| SpaceX | 55.94 | 54.80 |
| Tesla | 52.38 | 58.30 |
| Trump | 48.31 | 60.50 |
| Avg | 51.73 | 54.32 |

Table 9: Stance Detection Evaluation on MT-CDS Dataset w.r.t each class: USDC dataset in training and MT-CDS dataset in testing.

| Dataset | Against | Favor |
|---------|---------|-------|
| Biden | 34.40 | 58.80 |
| Bitcoin | 41.40 | 61.30 |
| SpaceX | 44.10 | 65.50 |
| Tesla | 49.0 | 67.50 |
| Trump | 54.5 | 66.4 |

### N.3 SLM FINETUNING: TRANSFER LEARNING PERFORMANCE ON TWITTER-STANCE DATASET

This dataset focuses on extracting stance (denying vs. supporting opinions) from Twitter posts, specifically targeting replies and quotes on controversial issues. It is tailored to the specific challenges of stance detection on Twitter, particularly in controversial and rumor-related contexts. This dataset consists of 5 classes such as Implicit denial, Explicit denial, Implicit support, Explicit support, and Quotes. These classes are similar to our USDC 5-class stance labels. Below are the accuracies we obtained on twitter-stance dataset. We also report individual class labels F1-score as follows: Denial (0.53), Support ( 0.32), Stance Not Inferrable (0.184). From Table 10 in Villa-Cox et al. (2020), we observe that the combined quotes and replies achieve a micro F1-score of 0.45, while our approach obtained a score of 0.43, which is close to the performance of human-annotated labels. Additionally, similar to Villa-Cox et al. (2020), our results show that the denial class performs better than the support class.

In conclusion, the results indicate that LLM-generated annotations of the USDC dataset are a viable alternative to human labels for stance detection tasks, demonstrating the substantial potential for automating and scaling up such complex annotation processes in long user conversation data.

Table 10: Stance Detection Evaluation on Twitter-stance Dataset w.r.t each class: USDC dataset in training and Twitter-stance dataset in testing.

| Dataset | Best Micro F1-score | USDC Micro F1-score |
|---------|---------------------|---------------------|
| Twitter-stance | 0.45 | 0.43 |

## O    INDIVIDUAL USER RESPONSES WITHIN THEIR SPECIFIC CONTEXT VS. ENTIRE CONVERSATION AT ONCE FOR STANCE AND DOGMATISM

For a given user, we consider each of their responses in the context of the topic and the comment they are responding to. We then use GPT-4 and Mistral-Large settings to assess annotations for the stance and dogmatism tasks. Using these generated annotations, we compare them to the annotations extracted from full-context conversations. The comparison statistics for stance and dogmatism tasks are reported in the Table 11 (Appendix).

The results from this experiment suggest that assessing each response individually within its context, and then aggregating the results, produces labels that are not identical to those derived from analyzing the entire conversation context. The higher percentage match with GPT-4 indicates that this method is fairly reliable. However, the differences in labels ( 30% with GPT-4 and  50% with Mistral-Large) highlight the importance of considering the full context for optimizing stance and dogmatism assessments.

Table 11: Individual user responses within their specific context vs. entire conversation at once for stance and dogmatism

| (a) Dogmatism Labels | | (b) Stance Labels | |
|----------------------|-----------------|-------------------|------------|
| **Comparison** | **Percentage Match** | **Comparison** | **Percentage** |
| GPT Labels Equal | 70.37% | GPT Labels Equal | 68.54% |
| GPT Labels Not Equal | 29.63% | GPT Labels Not Equal | 31.46% |
| Mistral Labels Equal | 53.70% | Mistral Labels Equal | 52.40% |
| Mistral Labels Not Equal | 46.30% | Mistral Labels Not Equal | 47.60% |

## P    INTER-ANNOTATOR AGREEMENT (IAA) BETWEEN HUMAN ANNOTATORS

We computed the Inter-Annotator Agreement (IAA) between human annotators as well. The Tables 12 and 13 report the IAA scores for both stance detection and dogmatism detection tasks among the human annotators.

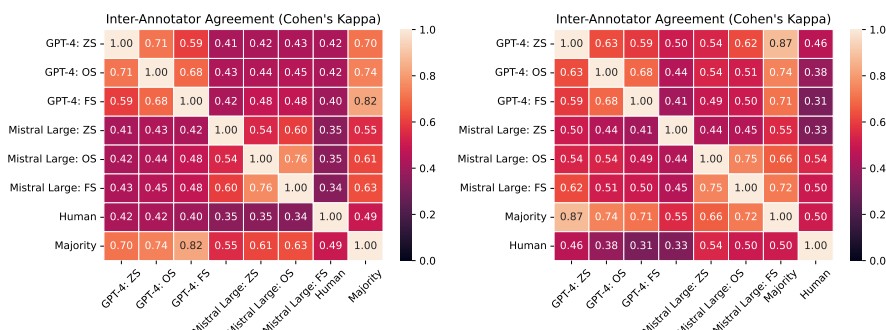

Figure 20: Inter-annotator agreement (IAA) on test dataset: Cohen's Kappa score across 8 settings: two different models (2 models×3 settings), majority voting and human annotations for Stance (left) and Dogmatism (right) tasks.

Table 12: Stance Detection

|        | Human1 | Human2 | Human3 |
|--------|--------|--------|--------|
| Human1 | 1.00   | 0.62   | 0.55   |
| Human2 | 0.62   | 1.00   | 0.57   |
| Human3 | 0.55   | 0.57   | 1.00   |

Table 13: Dogmatism Identification

|        | Human1 | Human2 | Human3 |
|--------|--------|--------|--------|
| Human1 | 1.00   | 0.57   | 0.51   |
| Human2 | 0.57   | 1.00   | 0.52   |
| Human3 | 0.51   | 0.52   | 1.00   |

## Q ROBUSTNESS ANALYSIS OF HUMAN-LLM ANNOTATIONS

Fig. 21 presents a heatmap comparing human-annotated labels and majority voting labels from LLMs, illustrating the class-specific agreement for Stance and Dogmatism tasks. From Fig. 21, we make the following observations for Stance classification task: (i) The "Stance Not Inferrable" (SNI) and "Strongly Against" (SGA) classes exhibit high agreement between human annotations and LLM predictions, as indicated by the strong diagonal values for these categories. (ii) "Somewhat in Favor" (SIF) and "Somewhat Against" (SOA) show substantial mismatches with human labels, leading to higher rates of false positives in LLM predictions. (iii) Notably, "Somewhat Against" (SOA) demonstrates the greatest level of disagreement, with frequent misclassification into neighboring categories such as "Strongly Against" (SGA) or "Somewhat in Favor" (SIF).

For Dogmatism task, we make following observations from Fig. 21 (right): (i) The "Firm but Open" (FBO) and "Open to Dialogue" (OTD) classes exhibit relatively high agreement, with strong diagonal values in the confusion matrix. These classes show better alignment between human labels and LLM predictions compared to other dogmatism categories. (ii) The "Deeply Rooted" (DR) and "Flexible" (FX) classes have significantly fewer samples and exhibit frequent misclassifications. For instance, "Deeply Rooted" (DR) is often misclassified as "Firm but Open" (FBO), indicating challenges in detecting extreme levels of dogmatism.

Overall, the significant mismatch for intermediate stance classes, particularly "Somewhat Against" in the stance detection task and "Open to Dialogue" in the dogmatism task, likely explains the moderate inter-annotator agreement (IAA) observed between human and LLM-generated labels.

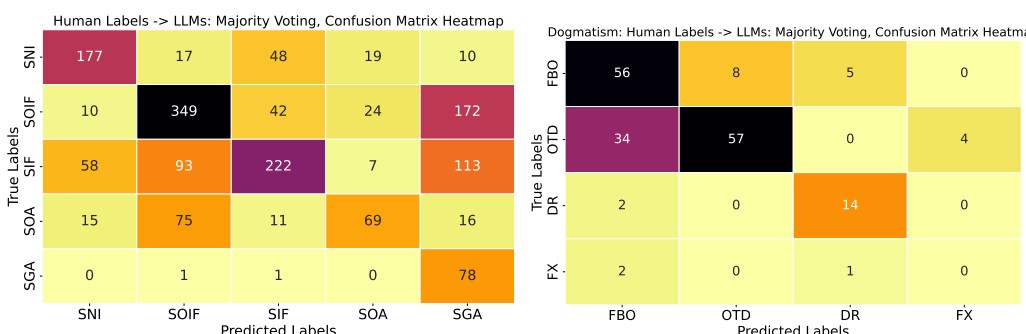

Figure 21: Confusion matrix between Human annotations and Majority voting labels of LLM annotations: (left) Stance Classification, (right) Dogmatism Identification.

## R  QUALITATIVE EXAMPLES DEMONSTRATING CASES WITH HIGH, MODERATE, AND LOW INTER-ANNOTATOR AGREEMENT

We now include qualitative examples demonstrating cases with high, moderate, and low inter-annotator agreement (IAA) for the Stance and Dogmatism tasks, as shown in Figs. R.1, R.2, R.3, R.4, R.5, R.6. In cases of high agreement, all LLMs consistently assign the same stance label to a user comment. For moderate agreement, some LLMs assign one stance class while others assign a neighboring stance class. For low agreement, GPT-4 assigns consistent stance labels across its three settings, but Mistral Large outputs differ for each setting.

### R.1  HIGH INTER-ANNOTATOR AGREEMENT STANCE EXAMPLES

```
{
    "submission_id": "abi4d2",
    "stance_id": "ed8f1x2",
    "stance_id_comment": "I'm not sure, but people like you speaking up
        helps. My new year's resolution is to promote what I think father
        's rights should be both here and in r/menslib. We don't get
        enough exposure and there are many misconceptions. But for your
        specific situation you could try r/legaladvice.",
    "reddit_link": "https://www.reddit.com/r/MensRights/comments/abi4d2/
        mens_issues_regarding_child_custody_and_child",

    "gpt41106preview_zero_shot_stance_label": "somewhat_in_favor",
    "gpt41106preview_one_shot_stance_label": "somewhat_in_favor",
    "gpt41106preview_few_shot_stance_label": "somewhat_in_favor",
    "mistrallargelatest_zero_shot_stance_label": "somewhat_in_favor",
    "mistrallargelatest_one_shot_stance_label": "somewhat_in_favor",
    "mistrallargelatest_few_shot_stance_label": "somewhat_in_favor",

    "gpt41106preview_zero_shot_stance_reason": "Author expresses intent
        to promote father's rights, showing support.",
    "gpt41106preview_one_shot_stance_reason": "Expresses a desire to
        promote father's rights, indicating support for change.",
    "gpt41106preview_few_shot_stance_reason": "Expresses a commitment to
        promoting father's rights, showing support for the cause.",
    "mistrallargelatest_zero_shot_stance_reason": "Author offers advice
        and expresses a desire to promote father's rights.",
    "mistrallargelatest_one_shot_stance_reason": "The author encourages
        someone to speak up about father's rights and offers advice.",
    "mistrallargelatest_few_shot_stance_reason": "Encourages speaking up
        for father's rights."
}
```

```
{
    "submission_id": "abt6bj",
    "stance_id": "ed2y40j",
    "stance_id_comment": "So many untested kits are by request from the
        \"victim\" and not due to discrimination or police failure",
    "reddit_link": "https://www.reddit.com/r/MensRights/comments/abt6bj/
        sane_sexual_assault_nurse_examiner_nurse_story",

    "gpt41106preview_zero_shot_stance_label": "somewhat_against",
    "gpt41106preview_one_shot_stance_label": "somewhat_against",
    "gpt41106preview_few_shot_stance_label": "somewhat_against",
    "mistrallargelatest_zero_shot_stance_label": "somewhat_against",
    "mistrallargelatest_one_shot_stance_label": "somewhat_against",
    "mistrallargelatest_few_shot_stance_label": "somewhat_against",

    "gpt41106preview_zero_shot_stance_reason": "Claims many untested kits
        are due to 'victim' requests.",
    "gpt41106preview_one_shot_stance_reason": "Claims many untested kits
        are due to victims' requests, showing some skepticism.",
    "gpt41106preview_few_shot_stance_reason": "Claims many untested kits
        are due to false accusers, showing distrust.",
    "mistrallargelatest_zero_shot_stance_reason": "Author claims that
        many untested rape kits are due to victim's request.",
    "mistrallargelatest_one_shot_stance_reason": "Implies that untested
        rape kits might be due to false accusations.",
    "mistrallargelatest_few_shot_stance_reason": "Believes many untested
        rape kits are due to false claims."
}

{
    "submission_id": "acnfv1",
    "stance_id": "acnfv1",
    "stance_id_comment": "The 2AF has received over $86,000 in donations
        just from Amazon Smile. That should be even higher! \n\nIf you
        haven't already, set them as who you donate to, and then **
        remember to checkout with smile.amazon.com, if you do regular
        amazon.com or the app, it won't donate anything.** \n\nIt will
        donate .5% of anything you purchase. That's $5 per every $1000
        you spend, which adds up fast if you use Amazon a lot. 2AF and
        Alan Gura won us *Heller* and continue to try and get gun cases
        to the Supreme Court. Help them keep doing so with a little of
        Amazon's money!",
    "reddit_link": "https://www.reddit.com/r/gunpolitics/comments/acnfv1/
        psa_set_your_amazon_smile_to_the_second_amendment",

    "gpt41106preview_zero_shot_stance_label": "strongly_in_favor",
    "gpt41106preview_one_shot_stance_label": "strongly_in_favor",
    "gpt41106preview_few_shot_stance_label": "strongly_in_favor",
    "mistrallargelatest_zero_shot_stance_label": "strongly_in_favor",
    "mistrallargelatest_one_shot_stance_label": "strongly_in_favor",
    "mistrallargelatest_few_shot_stance_label": "strongly_in_favor",

    "gpt41106preview_zero_shot_stance_reason": "Advocates for donations
        to 2AF, emphasizing its importance.",
    "gpt41106preview_one_shot_stance_reason": "Promotes 2AF donations
        through Amazon Smile, indicating strong support.",
    "gpt41106preview_few_shot_stance_reason": "Promotes Amazon Smile for
        2AF donations, showing strong support.",
    "mistrallargelatest_zero_shot_stance_reason": "Author promotes
        donating to 2AF through Amazon Smile, emphasizing its impact.",
    "mistrallargelatest_one_shot_stance_reason": "Promotes Amazon Smile
        donations to 2AF, emphasizing its impact.",
    "mistrallargelatest_few_shot_stance_reason": "Promotes donating to 2
        AF through Amazon Smile, highlighting its achievements."
}
```

```
{
    "submission_id": "abt6bj",
    "stance_id": "abt6bj",
    "stance_id_comment": "A female coworker of mine who is also a nurse
        was telling me that she took the SANE the class to become a nurse
         that assists patients in emergency rooms that claim they were
        raped. \nShe told me that if the dozens of patients she examined
        and collected evidence on their behalf, she only believed one of
        the \"victims\". \nThat one victim was a prostitute that was held
         against her will for 2 days repeatedly given heroin and raped. \
        nEvery other person she examined was highly suspicious for
        deception but she had to treat every single one like they were
        telling the truth. \nUltimately she quit because she felt she was
         just enabling women to file false reports against men, or
        feeding their desire to be victims or be manipulative. \nThese
        false accusers either need to be prosecuted or court ordered
        mental health services. This is not ok. It is why I rarely
        believe women when they say they were victims. It is unfortunate
        because rape is absolutely a horrible crime and should be
        punished severely when a criminal rapist is caught. But women
        need to stop lying. ",
    "reddit_link": "https://www.reddit.com/r/MensRights/comments/abt6bj/
        sane_sexual_assault_nurse_examiner_nurse_story",

    "gpt41106preview_zero_shot_stance_label": "strongly_against",
    "gpt41106preview_one_shot_stance_label": "strongly_against",
    "gpt41106preview_few_shot_stance_label": "strongly_against",
    "mistrallargelatest_zero_shot_stance_label": "strongly_against",
    "mistrallargelatest_one_shot_stance_label": "strongly_against",
    "mistrallargelatest_few_shot_stance_label": "strongly_against",

    "gpt41106preview_zero_shot_stance_reason": "Author expresses strong
        disbelief in most rape claims.",
    "gpt41106preview_one_shot_stance_reason": "Author expresses strong
        disbelief in most rape claims and suggests severe consequences
        for false accusers.",
    "gpt41106preview_few_shot_stance_reason": "Author expresses strong
        disbelief in most rape claims.",
    "mistrallargelatest_zero_shot_stance_reason": "Author expresses
        skepticism towards rape claims and believes most are false.",
    "mistrallargelatest_one_shot_stance_reason": "Expresses strong
        disbelief in rape claims and blames women for lying.",
    "mistrallargelatest_few_shot_stance_reason": "Expresses strong
        disbelief in rape claims and blames women for lying."
}
```

## R.2 MODERATE INTER-ANNOTATOR AGREEMENT STANCE EXAMPLES

```
{
    "submission_id": "abi4d2",
    "stance_id": "ed8gah6",
    "stance_id_comment": "No, people need to hear this. When they talk
        about \"best interest standards\" for kids, the top thing on that
         list should be not one, but two loving parents. If you have two
        people who want to be with their children how can you deny them
        that? Instead they push \"stable home\" meaning a preference for
        one parent over another. I have 50/50 custody and my kids are
        very well adjusted, and I would say more adaptable than other
        kids.",
    "reddit_link": "https://www.reddit.com/r/MensRights/comments/abi4d2/
        mens_issues_regarding_child_custody_and_child",
```

```
    "gpt41106preview_zero_shot_stance_label": "somewhat_in_favor",
    "gpt41106preview_one_shot_stance_label": "strongly_in_favor",
    "gpt41106preview_few_shot_stance_label": "somewhat_in_favor",
    "mistrallargelatest_zero_shot_stance_label": "strongly_in_favor",
    "mistrallargelatest_one_shot_stance_label": "somewhat_in_favor",
    "mistrallargelatest_few_shot_stance_label": "strongly_in_favor",

    "gpt41106preview_zero_shot_stance_reason": "Author argues for equal
        parenting, indicating a supportive stance.",
    "gpt41106preview_one_shot_stance_reason": "Strongly advocates for
        equal parenting, criticizing the current system.",
    "gpt41106preview_few_shot_stance_reason": "Advocates for equal
        parenting, criticizing the preference for one parent over another
        .",
    "mistrallargelatest_zero_shot_stance_reason": "Author advocates for
        equal custody rights for both parents.",
    "mistrallargelatest_one_shot_stance_reason": "The author advocates
        for children having two loving parents and shares their positive
        experience with 50/50 custody.",
    "mistrallargelatest_few_shot_stance_reason": "Advocates for equal
        custody and adaptability of children."
}

{
    "submission_id": "abp7hw",
    "stance_id": "ed3dbvc",
    "stance_id_comment": "Uh, yeah it's about attacking female privilege
        too.\n\nAnd yeah it's against \"SJW bullshit\" meaning anti-male
        prejudice and hate speech.\n\nI'm guessing the many women here
        would be surprised to learn any of that is attacking women.  Can
        I still criticize the KKK?  Or is that anti-White racism in your
        opinion?",
    "reddit_link": "https://www.reddit.com/r/MensRights/comments/abp7hw/
        a_reminder_that_this_sub_is_about_mens_rights_and",

    "gpt41106preview_zero_shot_stance_label": "strongly_in_favor",
    "gpt41106preview_one_shot_stance_label": "strongly_in_favor",
    "gpt41106preview_few_shot_stance_label": "strongly_against",
    "mistrallargelatest_zero_shot_stance_label": "strongly_in_favor",
    "mistrallargelatest_one_shot_stance_label": "strongly_against",
    "mistrallargelatest_few_shot_stance_label": "strongly_against",

    "gpt41106preview_zero_shot_stance_reason": "Author strongly supports
        attacking female privilege and anti-male prejudice.",
    "gpt41106preview_one_shot_stance_reason": "Asserts that attacking
        female privilege and opposing 'SJW bullshit' is part of the
        subreddit's purpose.",
    "gpt41106preview_few_shot_stance_reason": "Clearly opposes the idea
        that attacking female privilege is wrong and equates it with
        fighting prejudice.",
    "mistrallargelatest_zero_shot_stance_reason": "Author is in favor of
        attacking female privilege and criticizing the KKK.",
    "mistrallargelatest_one_shot_stance_reason": "Criticizes female
        privilege and SJWs, implying a negative stance towards them.",
    "mistrallargelatest_few_shot_stance_reason": "Criticizes female
        privilege and anti-male prejudice, implying a strong stance
        against them."
}
```

### R.3 LEAST INTER-ANNOTATOR AGREEMENT STANCE EXAMPLES

```
{
    "submission_id": "ach1kf",
```

```
    "stance_id": "ed83u16",
    "stance_id_comment": "It has nothing to do with logic it\u2019s all
        faith and emotion. I am always amazed that the atheist subs are
        so anti gun. ",
    "reddit_link": "https://www.reddit.com/r/gunpolitics/comments/ach1kf/
        a_easy_way_to_shutdown_anyone_who_brings_up",

    "gpt41106preview_zero_shot_stance_label": "somewhat_against",
    "gpt41106preview_one_shot_stance_label": "somewhat_against",
    "gpt41106preview_few_shot_stance_label": "somewhat_against",
    "mistrallargelatest_zero_shot_stance_label": "strongly_in_favor",
    "mistrallargelatest_one_shot_stance_label": "stance_not_inferrable",
    "mistrallargelatest_few_shot_stance_label": "somewhat_in_favor",

    "gpt41106preview_zero_shot_stance_reason": "Claims anti-gun sentiment
         is based on faith and emotion, indicating disagreement.",
    "gpt41106preview_one_shot_stance_reason": "Criticizes anti-gun
        sentiment as based on faith and emotion.",
    "gpt41106preview_few_shot_stance_reason": "Expresses a negative
        stance towards anti-gun views, attributing them to emotion rather
         than logic.",
    "mistrallargelatest_zero_shot_stance_reason": "Author dismisses anti-
        gun arguments as illogical.",
    "mistrallargelatest_one_shot_stance_reason": "Mentions atheist subs
        being anti-gun but does not express a clear stance.",
    "mistrallargelatest_few_shot_stance_reason": "Expresses disbelief
        towards anti-gun atheists, implying support for guns."
}

{
    "submission_id": "ach1kf",
    "stance_id": "edb7w5z",
    "stance_id_comment": "Sure but let\u2019s blame the tool as well,
        that is the point. Booze does not make someone a molester and a
        gun does not make someone a killer. ",
    "reddit_link": "https://www.reddit.com/r/gunpolitics/comments/ach1kf/
        a_easy_way_to_shutdown_anyone_who_brings_up",

    "gpt41106preview_zero_shot_stance_label": "somewhat_against",
    "gpt41106preview_one_shot_stance_label": "strongly_against",
    "gpt41106preview_few_shot_stance_label": "strongly_against",
    "mistrallargelatest_zero_shot_stance_label": "strongly_in_favor",
    "mistrallargelatest_one_shot_stance_label": "somewhat_against",
    "mistrallargelatest_few_shot_stance_label": "somewhat_in_favor",

    "gpt41106preview_zero_shot_stance_reason": "Argues against blaming
        tools (alcohol/guns) for actions, indicating stance.",
    "gpt41106preview_one_shot_stance_reason": "Strongly argues against
        blaming tools (guns) for actions of people.",
    "gpt41106preview_few_shot_stance_reason": "Clearly opposes the idea
        of blaming tools for actions, aligning with pro-gun stance.",
    "mistrallargelatest_zero_shot_stance_reason": "Author argues against
        blaming tools (guns) for crimes.",
    "mistrallargelatest_one_shot_stance_reason": "Suggests not blaming
        the tool (gun) but does not directly express a stance on gun
        control.",
    "mistrallargelatest_few_shot_stance_reason": "Suggests not blaming
        tools (guns) for the actions of people."
}
```

### R.4 HIGH INTER-ANNOTATOR AGREEMENT DOGMATISM EXAMPLES

```
{
```

```
    "submission_id": "abp7hw",
    "author_key": "author1",
    "author_key_name": "DavidByron2",
    "reddit_link": "https://www.reddit.com/r/MensRights/comments/abp7hw/
        a_reminder_that_this_sub_is_about_mens_rights_and",

    "gpt41106preview_zero_shot_dogmatism_label": "Firm but Open",
    "gpt41106preview_one_shot_dogmatism_label": "Firm but Open",
    "gpt41106preview_few_shot_dogmatism_label": "Firm but Open",
"mistrallargelatest_zero_shot_dogmatism_label": "Firm but Open",
"mistrallargelatest_one_shot_dogmatism_label": "Firm but Open",
"mistrallargelatest_few_shot_dogmatism_label": "Firm but Open",
    "gpt41106preview_zero_shot_dogmatism_reason": "Author holds strong
        views against SJWs but engages in discussion.",
    "gpt41106preview_one_shot_dogmatism_reason": "Strong views on the
        subreddit's purpose but engages without dismissing others.",
    "gpt41106preview_few_shot_dogmatism_reason": "Strongly opposes
        certain ideas but does not dismiss opposing views outright.",
    "mistrallargelatest_zero_shot_dogmatism_reason": "Author expresses
        strong opinions but also engages in discussion and asks questions
        .",
"mistrallargelatest_one_shot_dogmatism_reason": "Expresses strong
    opinions but engages in discussion and asks questions.",
"mistrallargelatest_few_shot_dogmatism_reason": "Expresses strong
    opinions but engages in discussion and asks questions, showing
    openness to dialogue."
}

{
    "submission_id": "abp7hw",
    "author_key": "author2",
    "author_key_name": "goodmod",
    "reddit_link": "https://www.reddit.com/r/MensRights/comments/abp7hw/
        a_reminder_that_this_sub_is_about_mens_rights_and",

    "gpt41106preview_zero_shot_dogmatism_label": "Open to Dialogue",
    "gpt41106preview_one_shot_dogmatism_label": "Open to Dialogue",
    "gpt41106preview_few_shot_dogmatism_label": "Open to Dialogue",
"mistrallargelatest_zero_shot_dogmatism_label": "Open to Dialogue",
"mistrallargelatest_one_shot_dogmatism_label": "Open to Dialogue",
"mistrallargelatest_few_shot_dogmatism_label": "Open to Dialogue",
    "gpt41106preview_zero_shot_dogmatism_reason": "Author enforces rules
        but is open to different opinions and dialogue.",
    "gpt41106preview_one_shot_dogmatism_reason": "Welcomes different
        opinions and upholds subreddit rules fairly.",
    "gpt41106preview_few_shot_dogmatism_reason": "Supports subreddit
        rules while acknowledging the validity of critical opinions.",
    "mistrallargelatest_zero_shot_dogmatism_reason": "Author is open to
        different viewpoints and engages in meaningful conversation.",
"mistrallargelatest_one_shot_dogmatism_reason": "Focuses on enforcing
    rules and addressing concerns without expressing personal opinions or
     dogmatic views.",
"mistrallargelatest_few_shot_dogmatism_reason": "Focuses on enforcing
    rules and engaging in conversation without expressing personal
    opinions on the topic."
}

{
    "submission_id": "acmcge",
    "author_key": "author1",
    "author_key_name": "Will_Power",
    "reddit_link": "https://www.reddit.com/r/climatechange/comments/
        acmcge/how_important_is_wind_power_and_what_is_its",

    "gpt41106preview_zero_shot_dogmatism_label": "Deeply Rooted",
```

```
    "gpt41106preview_one_shot_dogmatism_label": "Deeply Rooted",
    "gpt41106preview_few_shot_dogmatism_label": "Deeply Rooted",
"mistrallargelatest_zero_shot_dogmatism_label": "Deeply Rooted",
"mistrallargelatest_one_shot_dogmatism_label": "Deeply Rooted",
"mistrallargelatest_few_shot_dogmatism_label": "Deeply Rooted",
    "gpt41106preview_zero_shot_dogmatism_reason": "Consistently dismisses
        counterarguments, maintains strong stance without considering
        alternative views.",
    "gpt41106preview_one_shot_dogmatism_reason": "Consistently dismisses
        counterarguments and maintains a rigid stance.",
    "gpt41106preview_few_shot_dogmatism_reason": "Consistently argues
        against wind power with strong conviction, dismissing opposing
        views.",
    "mistrallargelatest_zero_shot_dogmatism_reason": "Author is firm in
        their stance against wind power and refuses to consider
        alternative viewpoints.",
"mistrallargelatest_one_shot_dogmatism_reason": "Shows a strong,
    unchangeable belief in the negative impact of wind power and is
    unlikely to be swayed.",
"mistrallargelatest_few_shot_dogmatism_reason": "Consistently maintains
    their stance against wind power and accuses the other author of being
     influenced by ideology."
}

{
    "submission_id": "brn0ti",
    "author_key": "author1",
    "author_key_name": "immibis",
    "reddit_link": "https://www.reddit.com/r/prolife/comments/brn0ti/
        what_do_prolifers_think_about_those_with_a_high",

    "gpt41106preview_zero_shot_dogmatism_label": "Flexible",
    "gpt41106preview_one_shot_dogmatism_label": "Flexible",
    "gpt41106preview_few_shot_dogmatism_label": "Flexible",
"mistrallargelatest_zero_shot_dogmatism_label": "Flexible",
"mistrallargelatest_one_shot_dogmatism_label": "Flexible",
"mistrallargelatest_few_shot_dogmatism_label": "Flexible",
    "gpt41106preview_zero_shot_dogmatism_reason": "Author's comments are
        off-topic, showing no strong beliefs.",
    "gpt41106preview_one_shot_dogmatism_reason": "The author's comments
        are unrelated to the topic, indicating a lack of strong opinion
        on the matter.",
    "gpt41106preview_few_shot_dogmatism_reason": "The author's comments
        are consistently off-topic, indicating a lack of engagement with
        the issue.",
"mistrallargelatest_zero_shot_dogmatism_reason": "The author does not
    express a clear stance on the topic and seems open to different
    interpretations.",
"mistrallargelatest_one_shot_dogmatism_reason": "The author does not
    express a clear stance or engage in meaningful discussion on the
    topic.",
"mistrallargelatest_few_shot_dogmatism_reason": "The author does not
    express a clear stance or opinion on the topic."
}
```

## R.5    MODERATE INTER-ANNOTATOR AGREEMENT DOGMATISM EXAMPLES

```
{
    "submission_id": "acqq1a",
    "author_key": "author1",
    "author_key_name": "Meyright",
    "reddit_link": "https://www.reddit.com/r/MensRights/comments/acqq1a/
        ive_noticed_a_worrying_trend_in_the_mensrights",
```

```
     "gpt41106preview_zero_shot_dogmatism_label": "Open to Dialogue",
     "gpt41106preview_one_shot_dogmatism_label": "Firm but Open",
     "gpt41106preview_few_shot_dogmatism_label": "Firm but Open",
"mistrallargelatest_zero_shot_dogmatism_label": "Firm but Open",
"mistrallargelatest_one_shot_dogmatism_label": "Open to Dialogue",
"mistrallargelatest_few_shot_dogmatism_label": "Open to Dialogue",
     "gpt41106preview_zero_shot_dogmatism_reason": "Engages with different
          viewpoints, acknowledges valid points, and calls for improvement
          .",
     "gpt41106preview_one_shot_dogmatism_reason": "Meyright shows a
         willingness to engage with different viewpoints but maintains a
         consistent perspective.",
     "gpt41106preview_few_shot_dogmatism_reason": "Shows a consistent
         stance against misrepresentation but remains open to evidence of
         discrimination.",
     "mistrallargelatest_zero_shot_dogmatism_reason": "Meyright shows a
         willingness to engage in discussion and consider different
         viewpoints, but holds firm on their beliefs.",
"mistrallargelatest_one_shot_dogmatism_reason": "Engages in discussion
    and provides nuanced perspectives without insisting on a single
    viewpoint.",
"mistrallargelatest_few_shot_dogmatism_reason": "Engages in thoughtful
    discussion, acknowledges different perspectives, and remains open to
    various viewpoints."
}

{
    "submission_id": "adp64t",
    "author_key": "author1",
    "author_key_name": "TBTop",
    "reddit_link": "https://www.reddit.com/r/gunpolitics/comments/adp64t/
        meaning_of_well_regulated_militia",

    "gpt41106preview_zero_shot_dogmatism_label": "Deeply Rooted",
    "gpt41106preview_one_shot_dogmatism_label": "Firm but Open",
    "gpt41106preview_few_shot_dogmatism_label": "Firm but Open",
"mistrallargelatest_zero_shot_dogmatism_label": "Deeply Rooted",
"mistrallargelatest_one_shot_dogmatism_label": "Firm but Open",
"mistrallargelatest_few_shot_dogmatism_label": "Deeply Rooted",
    "gpt41106preview_zero_shot_dogmatism_reason": "TBTop consistently
        argues with strong conviction, showing no openness to opposing
        interpretations of the 2nd amendment.",
    "gpt41106preview_one_shot_dogmatism_reason": "Author is steadfast in
        their pro-gun stance but engages in discussion.",
    "gpt41106preview_few_shot_dogmatism_reason": "Presents strong support
         for gun rights but engages in discussion and historical context
         .",
    "mistrallargelatest_zero_shot_dogmatism_reason": "Author consistently
         argues for their interpretation of the 2nd amendment and the
         right to bear arms, showing a strong, unchangeable belief.",
"mistrallargelatest_one_shot_dogmatism_reason": "Author presents strong
    opinions but engages in discussion and responds to counterarguments
    .",
"mistrallargelatest_few_shot_dogmatism_reason": "Consistently argues in
    favor of the right to bear arms, dismissing counterarguments and
    criticism."
}
```

## R.6    LEAST INTER-ANNOTATOR AGREEMENT DOGMATISM EXAMPLES

```
{
    "submission_id": "ar65mb",
    "author_key": "author2",
    "author_key_name": "CarolusMinimus",
```

```
    "reddit_link": "https://www.reddit.com/r/AntiVegan/comments/ar65mb/
        overtheinternet_has_been_suspended",

    "gpt41106preview_zero_shot_dogmatism_label": "Flexible",
    "gpt41106preview_one_shot_dogmatism_label": "Flexible",
    "gpt41106preview_few_shot_dogmatism_label": "Open to Dialogue",
"mistrallargelatest_zero_shot_dogmatism_label": "Deeply Rooted",
"mistrallargelatest_one_shot_dogmatism_label": "Firm but Open",
"mistrallargelatest_few_shot_dogmatism_label": "Deeply Rooted",
    "gpt41106preview_zero_shot_dogmatism_reason": "Limited data, but
        comments suggest a lack of strong dogmatic views.",
    "gpt41106preview_one_shot_dogmatism_reason": "Limited data, but shows
        no strong dogmatic tendencies in the available comments.",
    "gpt41106preview_few_shot_dogmatism_reason": "Limited data, but shows
        a negative stance without strong dogmatic language.",
"mistrallargelatest_zero_shot_dogmatism_reason": "Author expresses strong
    opinions and uses derogatory language, indicating a firm belief that
    is unlikely to be swayed.",
"mistrallargelatest_one_shot_dogmatism_reason": "Expresses opinions
    without being overly aggressive or dismissive of others.",
"mistrallargelatest_few_shot_dogmatism_reason": "Strongly negative
    opinions and unwillingness to engage in discussion."
}

{
    "submission_id": "bbzepe",
    "author_key": "author2",
    "author_key_name": "thesquarerootof1",
    "reddit_link": "https://www.reddit.com/r/AntiVegan/comments/bbzepe/
        im_an_exvegan_who_cant_stop_binge_eating_all_of",
    "gpt41106preview_zero_shot_dogmatism_label": "Deeply Rooted",
    "gpt41106preview_one_shot_dogmatism_label": "Firm but Open",
    "gpt41106preview_few_shot_dogmatism_label": "Firm but Open",
"mistrallargelatest_zero_shot_dogmatism_label": "Open to Dialogue",
"mistrallargelatest_one_shot_dogmatism_label": "Flexible",
"mistrallargelatest_few_shot_dogmatism_label": "Firm but Open",
"gpt41106preview_zero_shot_dogmatism_reason": "Uses strong language
    against veganism and diets, indicating deeply rooted beliefs.",
    "gpt41106preview_one_shot_dogmatism_reason": "Expresses strong views
        against veganism and diets but acknowledges moderation,
        indicating some openness.",
    "gpt41106preview_few_shot_dogmatism_reason": "Strong opinions on diet
        but shows some openness to moderation and clarification.",
"mistrallargelatest_zero_shot_dogmatism_reason": "Author engages in
    conversation and shares opinions without imposing them or dismissing
    others.",
"mistrallargelatest_one_shot_dogmatism_reason": "Encourages flexibility
    in diet choices and does not insist on a specific approach.",
"mistrallargelatest_few_shot_dogmatism_reason": "Expresses strong
    opinions on veganism but remains open to the idea of moderation."
}
```

## S  WEIGHTED COHEN'S KAPPA SCORE: IAA BETWEEN HUMAN LABELS AND LLM-GENERATED LABELS

We used the weighted Cohen's Kappa metric to compute the inter-annotator agreement (IAA) between human labels and LLM-generated labels across six settings, as well as majority voting, for the dogmatism task. Figure 22 reports the IAA on the test dataset, presenting the weighted Cohen's Kappa score across eight settings: two different models (2 models × 3 settings), majority voting, and human annotations for the dogmatism task. * This figure highlights that the weighted Cohen's

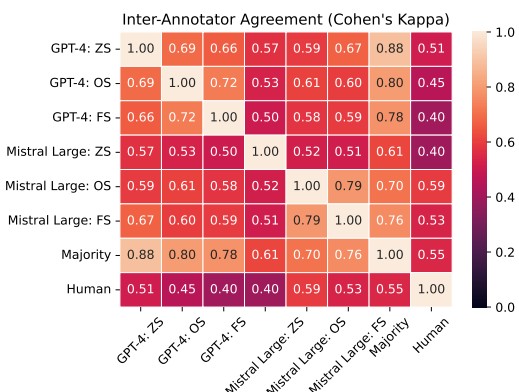

Figure 22: Inter-annotator agreement (IAA) on test dataset: Weighted Cohen's Kappa score across 8 settings: two different models (2 models×3 settings), majority voting and human annotations for the Dogmatism task.

Kappa metric improves the IAA between human annotations and the majority voting approach to 0.55, compared to the earlier score of 0.5 using the standard Cohen's Kappa metric. This indicates that the weighted Cohen's Kappa score effectively penalizes more distant disagreements, potentially leading to an improved measure of partial agreement.

