# OpenReview forum: "USDC: A Dataset of $\underline{U}$ser $\underline{S}$tance and $\underline{D}$ogmatism in Long $\underline{C}$onversations"
_ICLR.cc/2025/Conference — Submitted to ICLR 2025_

### Official Review · Reviewer_hG57 · 2024-10-29

**Soundness:** 3
**Presentation:** 3
**Contribution:** 2
**Rating:** 5
**Confidence:** 4

**Summary:**

Previous datasets on stance and dogmatism in user conversations are constructed at the post level. However, these datasets fail to capture fluctuations in users' opinions throughout entire conversational contexts. This paper introduces USDC, a dataset focused on user stance and dogmatism in long conversations. Inspired by the recent success of large language models (LLMs) in complex natural language processing tasks, the authors leverage Mistral Large and GPT-4 to annotate the training data for USDC. To ensure data quality, they manually labeled 200 test conversations for evaluation. They also conducted experiments to assess the performance of different models on the test set.

**Strengths:**

1. This paper introduces a novel dataset, USDC, which focuses on user stance and dogmatism in multi-user conversations.
2. It leverages large language models (LLMs) for efficient, scalable annotation of complex conversational data.
3. Extensive experiments are conducted using a variety of small language models.

**Weaknesses:**

In the experimental section of the paper, the authors often merely list the results without providing in-depth analysis of the underlying reasons. For more details, refer to the "Questions".

**Questions:**

Q1: Line 407 "2) For both tasks, the majority voting labels as ground truth has a relatively high performance, scoring above 50% weighted F1-score across several models." This expression is not very accurate. As shown in Table 1, for the Dogmatism Classification task, only 2 out of 14 models achieved an F1-score above 50% when using majority voting labels as ground truth. In contrast, when GPT-4 FS labels were used as ground truth, 11 out of 14 models outperformed the corresponding F1-scores obtained with majority voting labels. To enhance clarity and accuracy, I recommend that the authors revise these statements to reflect the results more precisely.

Q2: Line 410 "4) For GPT-4 annotations, in most cases, SLMs finetuned with few-shot annotations outperform those trained with zero and one-shot annotations. For Mistral Large annotations, SLMs finetuned with one-shot annotations perform the best." For the Mistral Large annotations, why do SLMs fine-tuned with one-shot annotations outperform those fine-tuned with corresponding few-shot annotations? Could the authors provide a hypothesis that  might explain this counterintuitive result?

---

> ### Author Response · Authors · 2024-11-18
>
> *We thank the reviewer for their strong positive, insightful and valuable comments and suggestions which are crucial for further strengthening our manuscript.*
>
> **Q1. I recommend that the authors revise these statements to reflect the results more precisely.**
>
> Thank you for pointing this out. We now revised the paragraph to better reflect the results as follows.
>
> * Correction of Majority Voting Performance for Dogmatism Task under SLM fine-tuning:
>   - Upon this clarification, for the Dogmatism Classification task, when using majority voting labels as ground truth, 2 out of 7 models in the fine-tuning setup achieve an F1-score above 50%. This distinction is now clearly reflected in the revised manuscript.
> * Correction for line 407: “For both tasks when finetuning, the majority voting labels as ground truth has a relatively high performance, scoring above 50\% weighted F1-score across several (7/7 for stance and 2/7 for dogmatism) models.”
>
> **Q2. why do SLMs fine-tuned with one-shot annotations outperform those fine-tuned with corresponding few-shot annotations? Could the authors provide a hypothesis that might explain this counterintuitive result?**
>
> Thank you for this question.
>
> * When fine-tuning with few-shot annotations, the model might overfit to the small number of examples provided. This overfitting can lead to poorer generalization to new data, whereas one-shot annotations might strike a better balance between learning and generalization.
> * For Mistral-large, we also observe this when computing inter-annotator agreement (IAA) with humans -- one-shot model has higher IAA compared to few-shot one.
>
> We believe that the large size of GPT4 helps it to avoid this problem.
>
> * Previous studies [Chen et al. 2023] have also shown that one-shot could be as good as or sometimes better compared to few-shot.
>
> [Chen et al. 2023]. "How Many Demonstrations Do You Need for In-context Learning?." EMNLP Findings 2023.

---

> > ### Author Response · Authors · 2024-11-23
> >
> > Dear Reviewer hG57,
> >
> > We appreciate your feedback and effort you have invested in evaluating our work.
> >
> > In response to your insightful comments, we have addressed the issues you highlighted. We believe these revisions significantly contribute to the clarity and completeness of the paper. We kindly request you to verify our response and consider updating your evaluation based on the revisions made.
> >
> > Should you have any further questions or suggestions, we are ready to provide additional information or clarification as needed.
> >
> > Thanks for your help

---

> > ### Comment · Reviewer_hG57 · 2024-11-25
> > **Response to Authors**
> >
> > Dear Authors,
> >
> > Thank you for your responses to my earlier comments. I appreciate the detailed clarifications and revisions provided. Below are my follow-up comments:
> >
> > Regarding Question 2, based on the results in Figure 13 of the paper, it appears that Mistral Large annotations in the few-shot setting achieve higher inter-annotator agreement with human annotations compared to the one-shot setting. This observation seems inconsistent with the second point mentioned in your response.
> >
> > Furthermore, Figure 13 also shows that GPT-4 consistently achieves higher inter-annotator agreement with human annotations across zero-shot, one-shot, and few-shot settings than Mistral Large.  In light of this, could majority voting based on the three GPT-4 settings further enhance the inter-annotator agreement with human annotations? Additionally, would models trained on the majority voting annotations derived from these GPT-4 settings achieve further enhanced performance on the test set (as shown in Table 1)?
> >
> > If I have misunderstood any aspect of the results or analyses, I welcome your clarification and additional insights. Thank you again for response!

---

> > > ### Author Response · Authors · 2024-11-25
> > >
> > > *We thank the reviewer for their valuable comments and suggestions which are crucial for further strengthening our manuscript.*
> > >
> > > **Regarding Q2:**
> > >
> > > Thanks for your observation.
> > >
> > > * Our conclusion in response to Q2, "For Mistral-large, we also observe this when computing inter-annotator agreement (IAA) with humans -- one-shot model has higher IAA compared to the few-shot one," was based on the overall test data for both stance and dogmatism tasks, as presented in Fig. 20. Specifically, for the stance detection task, the IAA is 0.35 for Mistral-large in the one-shot setting and 0.34 in the few-shot setting. Similarly, for the dogmatism detection task, the IAA is 0.54 for the one-shot setting and 0.50 for the few-shot setting.
> > > * However, as highlighted by the reviewer, Fig. 13 presents plots based on dividing the data into bins using timestamps. This experiment aimed to analyze the "lost in the middle" phenomenon. We agree with the reviewer that when analyzing the data in these bins, Mistral Large annotations in the few-shot setting achieve higher inter-annotator agreement with human annotations compared to the one-shot setting. This apparent inconsistency can be explained by Simpson's Paradox. While the overall test data indicates that κ(A,B)>κ(A,C), when examined within subsets (bins), the trend may reverse, showing κ(A,B)<κ(A,C). Simpson's Paradox occurs when trends present within individual groups disappear or reverse when these groups are combined.
> > > * We also conducted an analysis using the weighted Cohen's kappa score, as suggested by Reviewer WjM4, and is more appropriate for our setting. This metric accounts for the ordinal nature of the labels by assigning greater importance to closer agreements (e.g., 4 vs. 3) while proportionally penalizing more distant disagreements (e.g., 4 vs. 1). As shown in Fig. 22, the weighted Cohen's kappa scores are 0.59 for Mistral Large in the one-shot setting and 0.53 in the few-shot setting for the dogmatism task.
> > >
> > >
> > > **Could majority voting based on the three GPT-4 settings further enhance the inter-annotator agreement with human annotations?**
> > >
> > > Thank you for this question. We would like to clarify the process of obtaining labels in our USDC dataset and the role of both GPT-4 and Mistral Large models in the majority voting setup.
> > >
> > > **Label Distribution Across Models in the Majority Vote:**
> > > * For the stance detection task, 85% of the labels (7969 out of 9324) were obtained through majority voting involving both GPT-4 and Mistral models. The remaining 15% (1355 labels) were derived from GPT-4 in cases where there was a conflict.
> > > * Similarly, for the dogmatism task, 88% of the labels (1349 out of 1528) were obtained via majority voting, while the remaining 12% (179 labels) were assigned based on GPT-4’s output in conflict scenarios.
> > >
> > > **Why Both Models Are Necessary:**
> > > * The majority of the labels (more than 85% across tasks) were determined through collaboration between GPT-4 and Mistral, making both models integral to the labeling process.
> > > * GPT-4 is only used as a tie-breaker in the minority of cases (15% for stance detection and 12% for dogmatism), ensuring consistency and reliability for ambiguous cases without over-reliance on a single model.
> > >
> > > **Efficiency and Completeness:**
> > > This hybrid approach allows us to leverage the strengths of both models efficiently. Mistral provides cost-effective annotations for the majority of the data, while GPT-4 ensures high-quality resolution for edge cases, making the labeling process both scalable and accurate.
> > >
> > > We hope this explanation clarifies the necessity of both models in our dataset annotation process.

---

> > > > ### Author Response · Authors · 2024-11-26
> > > >
> > > > Dear Reviewer hG57,
> > > >
> > > > We appreciate your feedback and effort you have invested in evaluating our work.
> > > >
> > > > In response to your insightful comments, we have addressed the issues you highlighted. We believe these revisions significantly contribute to the clarity and completeness of the paper. We kindly request you to verify our response and consider updating your evaluation score based on the revisions made.
> > > >
> > > > Should you have any further questions or suggestions, we are ready to provide additional information or clarification as needed.
> > > >
> > > > Thanks for your help.

---

> > > > > ### Author Response · Authors · 2024-11-27
> > > > >
> > > > > Dear Reviewer hG57,
> > > > >
> > > > > Thank you for your feedback and for raising important ethical considerations regarding our paper. We would like to address them in detail.
> > > > >
> > > > > **Ethical Considerations:**
> > > > >
> > > > > * While Reddit posts and comments are publicly accessible (i.e., **the data is curated from public Reddit conversation threads**), and Reddit usernames are not real names, we want to clarify that we are not handling any personal demographic details of the users. **We only consider post IDs for mapping with users, ensuring that no user identity information is revealed in our research.** This approach helps to maintain user privacy while still allowing for meaningful analysis of the data.
> > > > >
> > > > > We hope this clarifies our approach and addresses the ethical concerns raised. We kindly request you to verify our previous and current responses and consider updating your evaluation score based on the revisions made.
> > > > >
> > > > > Should you have any further questions or suggestions, we are ready to provide additional information or clarification as needed.
> > > > >
> > > > > Regards,
> > > > >
> > > > > Authors

---

> > > > > > ### Author Response · Authors · 2024-11-27
> > > > > > **Looking forward to receiving your feedback**
> > > > > >
> > > > > > Dear Reviewer hG57,
> > > > > >
> > > > > > We truly understand the large workload that comes with reviewing and deeply appreciate the effort and time you have dedicated to reviewing our paper. As we are at the last stage of PDF updation, we want to kindly follow up to ensure there is sufficient time to address any remaining concerns you might have. Your recognition is highly important to us, and we sincerely hope to address all your concerns.
> > > > > >
> > > > > > Thank you once again for your efforts in reviewing our paper, and we look forward to receiving your feedback.
> > > > > >
> > > > > >
> > > > > > Regards,
> > > > > >
> > > > > > Authors

---

> > > > > > > ### Author Response · Authors · 2024-12-01
> > > > > > >
> > > > > > > Dear Reviewer hG57,
> > > > > > >
> > > > > > > As the author-reviewer discussion phase approaches its conclusion, we kindly request that you review our response and consider updating your evaluation score based on the revisions made.
> > > > > > >
> > > > > > > Should you have any further questions or suggestions, we are ready to provide additional information or clarification as needed.
> > > > > > >
> > > > > > > Regards,
> > > > > > >
> > > > > > > Authors

---

> > > > > > > > ### Comment · Reviewer_hG57 · 2024-12-03
> > > > > > > > **Response to Authors**
> > > > > > > >
> > > > > > > > Thank you for your responses. However, my concern regarding the majority voting with two models remains unresolved.I will maintain the current score.

---

> > > > > > > > > ### Author Response · Authors · 2024-12-03
> > > > > > > > >
> > > > > > > > > Dear Reviewer hG57,
> > > > > > > > >
> > > > > > > > > We appreciate your thoughtful feedback and acknowledge your concerns regarding the use of majority voting with two models.
> > > > > > > > >
> > > > > > > > > **How are conflicts resolved**
> > > > > > > > > * When generating annotations using both GPT-4 and Mistral, it’s possible that the two models might provide different annotations for the same conversation. To ensure consistency and accuracy in the final dataset, we have established a clear process for resolving these conflicts:
> > > > > > > > >
> > > > > > > > > **Majority Voting:**
> > > > > > > > > * What It Is: Majority voting is a method where, if multiple models or iterations are used, we look at all the annotations provided and choose the label that appears most frequently.
> > > > > > > > > * How It Helps: This approach helps reduce the impact of any potential error or bias from a single model. By relying on the most common label across models, we increase the likelihood that the chosen annotation is accurate.
> > > > > > > > >
> > > > > > > > > **Handling Situations with No Clear Majority:**
> > > > > > > > > * The Challenge: Sometimes, even with majority voting, the two models might provide different annotations, and neither label clearly dominates.
> > > > > > > > > * Our Solution: In these cases, we use the annotation provided by GPT-4 labels as the deciding factor or "gold standard."
> > > > > > > > > * Why GPT-4 labels?: We chose to prioritize GPT-4 annotations because human annotations have better IAA agreement with GPT-4 labels.  We have discussed label distribution across models in the majority vote in the previous responses.
> > > > > > > > >
> > > > > > > > > By following these steps, we aim to resolve conflicts in a way that enhances the reliability and accuracy of our dataset. We understand the importance of addressing potential conflicts thoroughly and believe that this method provides a balanced and effective solution.

---

> > > > > > > > > > ### Author Response · Authors · 2024-12-03
> > > > > > > > > >
> > > > > > > > > > Dear Reviewer hG57,
> > > > > > > > > >
> > > > > > > > > > As the author-reviewer discussion phase is set to conclude in the next 5 hours, we kindly invite you to review our response and consider revisiting your evaluation score in light of the additional information provided.
> > > > > > > > > >
> > > > > > > > > > Should you have any further questions or suggestions, we would be more than happy to offer any clarification or additional details to address your concerns.
> > > > > > > > > >
> > > > > > > > > > Thank you for your time and thoughtful consideration.
> > > > > > > > > >
> > > > > > > > > > Best regards,
> > > > > > > > > >
> > > > > > > > > > The Authors

---

### Official Review · Reviewer_sf39 · 2024-11-03

**Soundness:** 4
**Presentation:** 3
**Contribution:** 3
**Rating:** 6
**Confidence:** 3

**Summary:**

In this paper, the authors introduce a new dataset, named USDC.  It is as a dataset focusing on user stance and dogmatism in long conversations. Different from the previous methods based on labeling individual posts, this dataset can track changes in user opinions throughout the entire conversation, making it particularly useful for user personalization, market research, political campaigns, and content moderation. However, this paper lacks of more llms to be involved, only using Mistral Large and GPT-4, making the dataset has risk to be restricted by the knowledge in Mistral Large and GPT-4.

**Strengths:**

1. The article is well-written and easy for readers to understand.

2. It contributes a novel dataset, which is a unique collection focusing on user stance and dogmatism in long conversations.

3. Extensive experiments demonstrate that the annotations generated by LLMs are comparable to those generated by humans.

4. Fine-tuning and instruction-tuning multiple small language models and proved the effectiveness.

**Weaknesses:**

1. What I am particularly concerned about is that you only used two LLMs for data annotation, which poses a risk of missing knowledge from other regions and fields, especially knowledge that other models might possess.
2. I am also worried that in some cases, the system prompts may not be clear enough, leading to confusion in the large language models during annotation, such as inaccuracies in recognizing the author's stance.
3. Furthermore, the models face difficulties in identifying intermediate positions and ambiguous attitudes, which can easily lead to misunderstandings of user opinions. These issues will be even more pronounced in machine-annotated data, as you may not be able to provide sufficient information.

**Questions:**

Please use more experiments to convince me that the weaknesses not exist.

---

> ### Author Response · Authors · 2024-11-18
>
> *We thank the reviewer for their strong positive, insightful and valuable comments and suggestions which are crucial for further strengthening our manuscript.*
>
> **Q1. What I am particularly concerned about is that you only used two LLMs for data annotation, which poses a risk of missing knowledge from other regions and fields, especially knowledge that other models might possess.**
>
> Thank you for this question.
> * Before proceeding with LLM annotation using larger models, we first tested other versions of GPT, Mistral and LLaMA models, such as GPT-3.5 and Mistral-small and medium, LLaMA-2-70B. However, we found that these models failed to produce annotations in the desired format.
> * GPT4 is known to perform better than most other models across various regions and fields. Hence, we believe that a combination of GPT4 and mistral-large was good enough given our limited budget.
>
> **Q2. I am also worried that in some cases, the system prompts may not be clear enough, leading to confusion in the large language models during annotation, such as inaccuracies in recognizing the author's stance.**
>
> Thank you for this question.
>
> * The detailed system prompt for our LLM annotations is provided in Appendix C in the original paper.
> * It clearly lists the objective of the task, definition of stance, definition of dogmatism, detailed task description. Next we also provide detailed definitions of each of the stance labels as well as each of the document dogmatism labels.
> * Further we provide a detailed description of the different fields in the input data.
> * Lastly we have also provided helpful instructions for effective annotation where rather than just requesting the model to assign the label, we also asked it to provide a concise justification.
> * We therefore think that our system prompt is clear enough. Is there anything specific that you think we could have included?
>
> **Q3. Furthermore, the models face difficulties in identifying intermediate positions and ambiguous attitudes, which can easily lead to misunderstandings of user opinions. These issues will be even more pronounced in machine-annotated data, as you may not be able to provide sufficient information.**
>
> * For obtaining annotations, we provided the full conversation. Thus, we are able to provide sufficient information to the models (as much as we could provide to human annotators).
> * In the recency bias experiments (lines 477 to 485), we observe that providing full context to models is better than providing just the recent context i.e. prior context. Hence, we provided full context to get our final annotations.
> * Carefully designed system prompt, Few-shot prompting, majority voting are different ways in which we have attempted to obtain good quality annotations from LLMs.
> * Lastly, GPT4 and mistral have significantly improved abilities to understand and maintain context over long conversations.
>
> **Q4. Please use more experiments to convince me that the weaknesses not exist.**
>
> * For weakness 1, we have already argued that GPT4 and mistral can handle most regions and fields. Also, we have limited budget.
> * For weakness 2, we have already provided a clear system prompt.
> * For weakness 3, we have already designed robust methods to ensure accurate labels.
>   - We now include qualitative examples demonstrating cases with high, moderate, and low inter-annotator agreement (IAA) for the Stance and Dogmatism tasks, as shown in Appendix Figs. R.1, R.2, R.3, R.4, R.5 and R.6. In cases of high agreement, all LLMs consistently assign the same stance label to a user comment. For moderate agreement, some LLMs assign one stance class while others assign a neighboring stance class. For low agreement, GPT-4 assigns consistent stance labels across its three settings, but Mistral Large outputs differ for each setting. We have added these examples to Appendix R in the revised manuscript to illustrate where the labeling approach is most, moderate and least effective.

---

> > ### Author Response · Authors · 2024-11-23
> >
> > Dear Reviewer sf39,
> >
> > We appreciate your feedback and effort you have invested in evaluating our work.
> >
> > In response to your insightful comments, we have addressed the issues you highlighted. We believe these revisions significantly contribute to the clarity and completeness of the paper. We kindly request you to verify our response and consider updating your evaluation based on the revisions made.
> >
> > Should you have any further questions or suggestions, we are ready to provide additional information or clarification as needed.
> >
> > Thanks for your help

---

> > > ### Author Response · Authors · 2024-11-26
> > >
> > > Dear Reviewer sf39,
> > >
> > > Thank you for your valuable and constructive comments, which have significantly enhanced the quality of our manuscript.
> > >
> > > We have carefully addressed the comments and questions you raised, making significant revisions to enhance the clarity and completeness of the paper. We kindly request you to verify our response and consider updating your evaluation score based on the revisions made.
> > >
> > > Should you have any further questions or suggestions, we are ready to provide additional information or clarification as needed.
> > >
> > > Thanks for your help.

---

> > > > ### Author Response · Authors · 2024-12-03
> > > >
> > > > Dear Reviewer sf39,
> > > >
> > > > We sincerely appreciate the time and effort you have dedicated to reviewing our paper.
> > > >
> > > > Since there are only a few hours remaining for reviewers to post messages to authors, we kindly request you to verify our response and consider updating your evaluation score based on the revisions made.
> > > >
> > > > Should you have any further questions or suggestions, we are ready to provide additional information or clarification as needed.
> > > >
> > > > Thanks for your help.

---

### Official Review · Reviewer_QAKS · 2024-11-03

**Soundness:** 3
**Presentation:** 3
**Contribution:** 3
**Rating:** 8
**Confidence:** 3

**Summary:**

It is very costly to have humans annotate Reddit threads with multiple posts to label the stances and dogmatism of multiple users.
Previous work considers posts independently, however, that is not the nature of the interaction.

This paper looks at if LLMs can encapsulate the nuances to understand user opinions and whether their opinions shift through the conversation.

LLMs are used to classify (1) user stances and (2) user dogmatism and express their reasoning for their classification. Every sample is annotated six times (two models x zero-shot, one-shot, and few-shot) and then the majority vote is taken.

The paper introduces the USDC dataset that includes 1528 dogmatism samples (user-level) and 9618 stance samples. It is able to capture contextual and opinion shifts. As such it can be used as an instruction-tuning dataset or evaluation benchmark. The authors also instruction-tune and fine-tune LLMs on the dataset.

**Strengths:**

1. The paper is well-motivated and easy to read.
2. The approach is straightforward and makes sense.
3. The added qualitative analysis is very important and nice to read.

**Weaknesses:**

1. The majority voting conflict makes me wonder why Mistral is used at all if, in cases of conflict, the decision maker is GPT4 (which is quite a costly model)?
2. Majority voting labels are used as ground-truth. It would be good to add experiments on what would happen if we train on unaggregated labels, as subjectivity is important in such a task.

**Questions:**

Why is it the case that instruction-tuning is better for stance and fine-tuning better for dogmatism?

---

> ### Author Response · Authors · 2024-11-18
>
> *We thank the reviewer for their strong positive, insightful and valuable comments and suggestions which are crucial for further strengthening our manuscript.*
>
> **Q1. The majority voting conflict makes me wonder why Mistral is used at all if, in cases of conflict, the decision maker is GPT4 (which is quite a costly model)?**
>
> Thank you for this question.  We would like to clarify the process of obtaining labels in our USDC dataset and the role of both GPT-4 and Mistral Large models in the majority voting setup.
>
> * Label Distribution Across Models:
>   - For the stance detection task, 85% of the labels (7969 out of 9324) were obtained through majority voting involving both GPT-4 and Mistral models. The remaining 15% (1355 labels) were derived from GPT-4 in cases where there was a conflict.
>   - Similarly, for the dogmatism task, 88% of the labels (1349 out of 1528) were obtained via majority voting, while the remaining 12% (179 labels) were assigned based on GPT-4’s output in conflict scenarios.
>
> * Why Both Models Are Necessary:
>   - The majority of the labels (more than 85% across tasks) were determined through collaboration between GPT-4 and Mistral, making both models integral to the labeling process.
>   - GPT-4 is only used as a tie-breaker in the minority of cases (15% for stance detection and 12% for dogmatism), ensuring consistency and reliability for ambiguous cases without over-reliance on a single model.
>
> * Efficiency and Completeness:
>   - This hybrid approach allows us to leverage the strengths of both models efficiently. Mistral provides cost-effective annotations for the majority of the data, while GPT-4 ensures high-quality resolution for edge cases, making the labeling process both scalable and accurate.
>
> We hope this explanation clarifies the necessity of both models in our dataset annotation process.
>
> **Q2. Majority voting labels are used as ground-truth. It would be good to add experiments on what would happen if we train on unaggregated labels, as subjectivity is important in such a task.**
>
> Thank you for this question.
> * We would like to clarify that our experiments are not limited to majority voting as ground truth. We have also performed fine-tuning and instruction tuning using zero-shot, one-shot, and few-shot labels from both GPT-4 and Mistral Large models as ground truth. This additional analysis provides a comprehensive evaluation of how different types of labels, including individual model labels, affect model performance.
>
> * Analysis of Majority Voting vs. Individual Model Labels:
>   - The results (Table 1 in main paper) demonstrate that majority voting labels consistently result in better model performance compared to labels obtained from individual models (e.g., GPT-4 or Mistral alone).
>   - Majority voting helps mitigate noise and outlier annotations by capturing a more robust consensus, improving generalization and reducing variability during training.
> * Unaggregated Labels Under Individual Model Performance:
>   - The analysis of unaggregated labels is equivalent to conducting experiments with individual model labels. The performance using GPT-4 or Mistral labels alone reflects the impact of unaggregated, model-specific subjectivity.
>   - These experiments highlight that while individual models capture useful task-specific features, aggregated majority voting labels lead to better alignment with the task’s ground truth.
>
> In summary, our study already incorporates an analysis of unaggregated labels via experiments on individual model labels.
>
> **Q3. Why is it the case that instruction-tuning is better for stance and fine-tuning better for dogmatism?**
>
> Thank you for this thoughtful question. Below, we clarify why fine-tuning appears to outperform instruction-tuning for the dogmatism detection task and provide additional analysis to support our findings.
>
> * Since dogmatism detection is inherently more complex and varied than stance detection, the model might struggle to generalize from the instructional data
> * We report the confusion matrix for dogmatism detection task in Fig. 9 in the Appendix.
>   - It shows significant misclassifications, especially for the ''Deeply Rooted'' and ''Flexible'' labels, with zero accuracy and F1-scores.
>   - On the other hand, the model performs moderately better for ''Firm but Open'' and ''Open to Dialogue'' classes with accuracies of 48.7\% and 64.4\%, respectively.
>   - The confusion matrix also indicates substantial confusion to distinguish between intermediate levels of dogmatism, such as ''Firm but Open'' and ''Open to Dialogue''.
>   - This analysis demonstrates that dogmatism detection demands robust handling of long-term dependencies and task-specific adaptations, which fine-tuning achieves more effectively than instruction-tuning.

---

> > ### Comment · Reviewer_QAKS · 2024-11-25
> >
> > Thank you for addressing my concerns. As I was already positive about this paper, I would like to keep my score the same!

---

> > > ### Author Response · Authors · 2024-11-25
> > >
> > > We appreciate the reviewer's positive feedback and are confident that it has enhanced the paper's quality.

---

### Official Review · Reviewer_WjM4 · 2024-11-05

**Soundness:** 3
**Presentation:** 2
**Contribution:** 3
**Rating:** 6
**Confidence:** 4

**Summary:**

This paper introduces a dataset for detecting stance and dogmatism in each turn of a conversation.
The dataset is based on Reddit conversations, where each turn is a reply within a thread.
To label the stance and dogmatism of each speaker—a task with inherent subjectivity—the authors use multiple prompting schemes across different models, finalizing each label through majority voting.

**Strengths:**

The dataset provides valuable insights into stance and dogmatic expressions in Reddit conversations, contributing a unique resource for analyzing opinion and belief expression in online discourse.

**Weaknesses:**

1. Moderate Inter-Annotator Agreement: The inter-annotator agreement between human and LLM annotations could be improved.

2. Fragmented Conversations: By selecting only the top two authors’ comments, the dataset lacks conversational continuity. The two selected authors’ comments are scattered across the thread, rather than forming a cohesive conversation.

**Questions:**

1. The observed inter-annotator agreement between LLM-human and human-human highlights the subjectivity of this task. If these labels are treated as ground truth in experiments, might this reduce the robustness of the results?

2. Are there any qualitative examples demonstrating cases with high, moderate, and low inter-annotator agreement? These could provide insight into where the labeling approach is most and least effective.

---

> ### Author Response · Authors · 2024-11-18
>
> *We thank the reviewer for their valuable comments and suggestions which are crucial for further strengthening our manuscript.*
>
> **Q1. Moderate Inter-Annotator Agreement: The inter-annotator agreement between human and LLM annotations could be improved.**
>
> Thank you for this question.
>
> * Typical social media based datasets especially those involving long text are difficult to label objectively. That said, an inter-annotator agreement 0.49 for stance and 0.50 for dogmatism is quite reasonable. Previous studies have reported similar agreement values *[Fast & Horvitz 2016] [Sakketou et al. 2022]*.
> * Possible ways to improve IAA is to ensure that the annotation guidelines are as objective as possible. To ensure this, we already performed several iterations (both manually as well as using prompt optimization methods) and have carefully designed the annotation guidelines. We have used the same guidelines as part of the prompt. The prompts are already included in the Appendix.
>
> *[Fast and Horvitz 2016], "Identifying dogmatism in social media: Signals and models." EMNLP (2016)*
>
> *[Sakketou et al. 2022], "Investigating user radicalization: A novel dataset for identifying fine-grained temporal shifts in opinion." arXiv preprint arXiv:2204.10190 (2022). https://arxiv.org/pdf/2204.10190*
>
> **Q2. Fragmented Conversations: By selecting only the top two authors’ comments, the dataset lacks conversational continuity. The two selected authors’ comments are scattered across the thread, rather than forming a cohesive conversation.**
>
> Thank you for raising this important question.
> * We would like to clarify that although we annotated stance and dogmatism labels for the top two authors, we included the full conversation thread with all user comments in our dataset. This ensures that there are no fragmented conversations, as the full context of every user comment is provided.
> * The dataset captures the entire thread, allowing the LLM to analyze how a user changes their opinion based on other users' comments within the context of the target conversation. This setup enables us to study how a user’s opinion evolves in response to other users’ comments throughout the entire conversation.
>
> * Our decision to focus on the two most active users stems from the observation that users with fewer comments often do not provide sufficient data to accurately assess their stance or dogmatism. Many users contribute only one or two comments, which is typically insufficient for determining their overall opinion or dogmatic nature.
> * By prioritizing the two most active users—who contribute approximately 50% of the comments in each conversation—we ensure that our analysis captures meaningful opinion fluctuations and provides a robust evaluation of stance and dogmatism. Opinion fluctuations, as well as stance and dogmatism detection can be useful to build moderation tools, which will be most applicable for active users.
> * While we acknowledge that including additional users could enhance the dataset’s completeness, we believe our current approach strikes a balance between computational feasibility, analytical depth and practical usability. We appreciate the reviewer’s insight and will clarify this aspect further in the revised manuscript.
>
> **Q3. Are there any qualitative examples demonstrating cases with high, moderate, and low inter-annotator agreement? These could provide insight into where the labeling approach is most and least effective.**
>
> Thank you for this question.
> * Based on the reviewer’s suggestion, we now include qualitative examples demonstrating cases with high, moderate, and low inter-annotator agreement (IAA) for the Stance and Dogmatism tasks, as shown in Appendix Figs. R.1, R.2, R.3, R.4, R.5 and R.6.
>   - In cases of high agreement, all LLMs consistently assign the same stance label to a user comment.
>   - For moderate agreement, some LLMs assign one stance class while others assign a neighboring stance class.
>   - For low agreement, GPT-4 assigns consistent stance labels across its three settings, but Mistral Large outputs differ for each setting.
> * We have added these examples to Appendix R in the revised manuscript to illustrate where the labeling approach is most, moderate and least effective.

---

> ### Author Response · Authors · 2024-11-18
>
> **Q4. The observed inter-annotator agreement between LLM-human and human-human highlights the subjectivity of this task. If these labels are treated as ground truth in experiments, might this reduce the robustness of the results?**
>
> * Typical social media based datasets especially those involving long text are difficult to label objectively. That said, an inter-annotator agreement 0.49 for stance and 0.50 for dogmatism is quite reasonable. Previous studies have reported similar agreement values *[Fast & Horvitz 2016] [Sakketou et al. 2022]*.
> * Possible ways to improve IAA is to ensure that the annotation guidelines are as objective as possible. To ensure this, we already performed several iterations (both manually as well as using prompt optimization methods) and have carefully designed the annotation guidelines. We have used the same guidelines as part of the prompt. The prompts are already included in the Appendix.
> * Using this dataset, we train llama-based models. These models are known to be robust to training label noise.
> * The IAA is around 0.5 but it only captures mismatches and not the degree of mismatch. In our case, the classes have a ranking order. The predictions from human vs LLM are close to each other in this order even when they mismatch.
>
> * Appendix Q Fig. 21 presents a heatmap comparing human-annotated labels and majority voting labels from LLMs, illustrating the class-specific agreement for Stance and Dogmatism tasks. From  Fig. 21 (left), we make the following observations:
>     - (i) The ''Stance Not Inferrable'' (SNI) and ''Strongly Against'' (SGA) classes exhibit high agreement between human annotations and LLM predictions, as indicated by the strong diagonal values for these categories. These classes achieve recall values of 0.97 and 0.65, respectively, when compared to human labels.
>     - (ii) ''Somewhat in Favor'' (SIF) and ''Somewhat Against'' (SOA) show substantial mismatches with human labels, leading to higher rates of false positives in LLM predictions.
>     - (iii) Notably, ''Somewhat Against'' (SOA) demonstrates the greatest level of disagreement (recall of 0.37), with frequent misclassification into neighboring categories such as ''Strongly Against'' (SGA) or ''Somewhat in Favor'' (SIF).
> * For Dogmatism task, we make following observations from Fig. 21 (right):
>     - (i) The ''Firm but Open'' (FBO) and ''Open to Dialogue'' (OTD) classes exhibit relatively high agreement, with strong diagonal values in the confusion matrix. These classes show better alignment between human labels and LLM predictions compared to other dogmatism categories.
>     - (ii) The ''Deeply Rooted'' (DR) and ''Flexible'' (FX) classes have significantly fewer samples and exhibit frequent misclassifications. For instance, ''Deeply Rooted’’ (DR) is often misclassified as ''Firm but Open’’ (FBO), indicating challenges in detecting extreme levels of dogmatism.
> * Overall, the significant mismatch for intermediate stance classes, particularly ''Somewhat Against’’ in the stance detection task and ''Open to Dialogue’’ in the dogmatism task, likely explains the moderate inter-annotator agreement (IAA) observed between human and LLM-generated labels.

---

> > ### Comment · Reviewer_WjM4 · 2024-11-20
> > **Response to Author**
> >
> > Dear Author,
> >
> > Thank you for your clarification and for including inter-annotator annotation examples in the Appendix.
> > This addition has helped me better understand the dataset.
> >
> > It seems that the labels are on an ordinal scale (e.g., "Deeply Rooted" = 4, "Firm but Open" = 3, etc.). Given this, it would make more sense to evaluate inter-annotator agreement using a weighted kappa metric, which accounts for the ordinal nature of the labels. (i.e., assigns greater importance to closer agreements (e.g., 4 vs. 3) and penalizes more distant disagreements (e.g., 4 vs. 1) proportionally.)
> >
> > I still believe that the asynchronous nature of the conversation between the two parties may not be sufficient for properly evaluating the user's stance and dogmatism within a cohesive conversational flow. This limitation could impact the accuracy and consistency of the evaluation. It would be helpful if you could clarify this aspect in your paper, perhaps by addressing how the asynchronous setup accounts for or mitigates this issue. Providing a discussion or justification for this design choice would strengthen the validity of the approach and make it clearer to readers.

---

> > > ### Author Response · Authors · 2024-11-20
> > >
> > > Dear Reviewer WjM4,
> > >
> > > We appreciate the reviewer’s positive feedback and are confident that it has contributed to enhancing the quality of our paper.
> > >
> > > We acknowledge the two points raised regarding the use of a weighted kappa metric and the asynchronous nature of the conversations. We will address these points promptly during this discussion period.
> > >
> > > Regards,
> > > The Authors

---

> > > > ### Author Response · Authors · 2024-11-23
> > > >
> > > > Dear Reviewer WjM4,
> > > >
> > > > We appreciate the reviewer’s positive feedback and are confident that it has contributed to enhancing the quality of our paper.
> > > >
> > > > * As per your suggestion, we used the weighted Cohen's Kappa metric to compute the inter-annotator agreement (IAA) between human labels and LLM-generated labels across six settings, as well as majority voting, for the dogmatism task. Appendix S, Figure 22 reports the IAA on the test dataset, presenting the weighted Cohen’s Kappa score across eight settings: two different models (2 models × 3 settings), majority voting, and human annotations for the dogmatism task.
> > > > * This figure highlights that the weighted Cohen’s Kappa metric improves the IAA between human annotations and the majority voting approach to 0.55, compared to the earlier score of 0.5 using the standard Cohen’s Kappa metric. This indicates that the weighted Cohen’s Kappa score effectively penalizes more distant disagreements, potentially leading to an improved measure of partial agreement.
> > > >
> > > > **Asynchronous Nature of User Conversations:**
> > > >
> > > > * We agree with the reviewer that Reddit conversations can indeed occur in real-time, but they predominantly unfold asynchronously. While users can respond immediately, delays between comments are common as users engage at different times. To address this, **our dataset includes the entire conversation thread with messages from all users**, arranged in the order of posts by timestamp **(not just the utterances of the 2 users)**. This ensures that all comments, regardless of posting time, are considered, preserving the context and continuity necessary for accurately evaluating stance and dogmatism. Our methods are designed to account for temporal gaps and contextual shifts, enabling robust analysis despite the asynchronous nature of these interactions.
> > > > * Also, while synchronous mechanisms like Facebook chat or Team chat could also be useful in evaluating stance and dogmatism, we believe that asynchronous communication is typically more thought-through because the users take time to think, form an opinion, think about consequences and then respond. Hence, evaluating user stance or dogmatism from asynchronous communication threads (while retaining entire conversation context) may be more useful than evaluating using synchronous real-time chat messengers.
> > > >
> > > > We hope these updates address your concerns and demonstrate our commitment to improving the manuscript. Thank you again for your insightful feedback.

---

### Author Response · Authors · 2024-11-18
**Summary of our responses and revision**

*We are grateful to all reviewers for their strong positive feedback, time and their constructive suggestions, which will further strengthen the impact of our work.*

**Summary of Reviewer Strengths:**

1. The paper is well-written, clear, and effectively motivated **(QAKS, sf39)**.
2. This paper introduces a novel contribution through the USDC dataset, which focuses on user stance and dogmatism in multi-user conversations **(WjM4, QAKS, sf39, hG57)**.
3. The methodology is straightforward and includes qualitative analysis, which adds depth to the findings **(QAKS, sf39)**
4. Extensive experiments, including fine-tuning and instruction-tuning of multiple small language models, validate the effectiveness of the proposed approach **(QAKS, sf39, hG57)**.
5. The work leverages large language models (LLMs) for efficient and scalable annotation, demonstrating that LLM-generated annotations are comparable to human annotations **(hG57)**.

**Additional changes to the draft during the rebuttal process**

We have updated the main manuscript and the appendix to address these following comments. The changes made in the manuscript are highlighted in blue color. The major additional changes are listed below.

1. **Qualitative examples demonstrating cases with high, moderate, and low inter-annotator agreement** (Reviewer WjM4, sf39): We have included qualitative examples in Appendix R of the revised manuscript, demonstrating cases with high, moderate, and low inter-annotator agreement (IAA) for the Stance and Dogmatism tasks, as illustrated in Figs. R.1–R.6. These examples highlight where the labeling approach is most effective, moderately effective, and least effective, showcasing consistent stance labels in high-agreement cases and discrepancies, particularly with Mistral Large, in low-agreement cases.

2. **Robustness analysis of Human-LLM Annotations** (Reviewer WjM4): We have included a heatmap in Appendix Q, Fig. 21, comparing human-annotated labels with majority voting labels from LLMs, highlighting class-specific agreement for the Stance and Dogmatism tasks. The significant mismatch in intermediate stance classes, particularly "Somewhat Against" in stance detection and "Open to Dialogue" in dogmatism, likely accounts for the moderate inter-annotator agreement (IAA) observed between human and LLM-generated labels.

3. **Weighted Cohen's Kappa score: IAA between human labels and LLM-generated labels** (Reviewer WjM4): Appendix S, Fig 22 presents weighted Cohen’s Kappa across eight settings, and highlights that the weighted Cohen’s Kappa metric improves the IAA between human annotations and the majority voting approach to 0.55.

We hope these revisions will satisfactorily address the concerns raised by the reviewers and elevate the overall quality of our work.

---

### Meta-Review · Area_Chair_kF15 · 2024-12-20

**Metareview:**

This work investigates Stance and Dogmatism in conversations, by using LLMs to automate the human annotation process on user stance classification and user dogmatism classification over 764 multi-user Reddit conversations. As a result, this work constructs the USDC dataset. Specifically, 200 test conversations are annotated by human.

Strengths:
1. The paper is well-written, clear, and well-motivated.
2. The paper constructs a new dataset for studying Stance and Dogmatism in conversations.
3. Experimental results validate the effectiveness of the proposed approach.

Weaknesses:
1. The reliability of the constructed dataset is questioned by all the reviewers, including the subjectivity (moderate inter-annotator agreement/majority vote as ground truth), unnatural conversations (fragmented discussion thread), only two LLMs for annotations, etc.
2. The experimental results lack in-depth analysis. Some are solved during the rebuttal, but some are still not satisfied by the reviewer.
3. There are only 200 human-annotated test conversations, which may not be sufficient for robust evaluation.

Overall, as a resource paper, the quality and reliability of the contributed dataset should be the key. However, according to the comments from the reviewers, despite the good quality of the writing and motivation of the problem, there are still many concerns on the curated dataset itself. Therefore, it would be better to further enhance the quality validation and even the size and diversity of the created dataset.

**Additional Comments On Reviewer Discussion:**

Reviewer QAKS remains positive towards this work, but the concerns from Reviewer WjM4 and hG57 are not fully addressed.

---

### Decision · Program_Chairs · 2025-01-22

Reject